# PACE: A Parallelizable Computation Encoder for Directed Acyclic Graphs

## Abstract

Optimization of directed acyclic graph (DAG) structures has many applications, such as neural architecture search (NAS) and probabilistic graphical model learning. Encoding DAGs into real vectors is a dominant component in most neural-network-based DAG optimization frameworks. Currently, most popular DAG encoders use an asynchronous message passing scheme which sequentially processes nodes according to the dependency between nodes in a DAG. That is, a node must not be processed until all its predecessors are processed. As a result, they are inherently not parallelizable. In this work, we propose a Parallelizable Attention-based Computation structure Encoder (PACE) that processes nodes simultaneously and encodes DAGs in parallel. We demonstrate the superiority of PACE through encoder-dependent optimization subroutines that search the optimal DAG structure based on the learned DAG embeddings. Experiments show that PACE not only improves the effectiveness over previous sequential DAG encoders with a significantly boosted training and inference speed, but also generates smooth latent (DAG encoding) spaces that are beneficial to downstream optimization subroutines.

## 1 Introduction

Directed acyclic graphs (DAGs) are ubiquitous in various real-world problems including neural architecture search (Elsken et al., 2019; Wen et al., 2020), source code modeling (Allamanis et al., 2018), structure learning of Bayesian networks (Koller & Friedman, 2009; Zhang et al., 2019), etc. One key challenge in DAG optimization problems is that it is not easy to use gradient strategies to quickly adjust the structure of a DAG towards the right direction due to the absence of gradient information. Some earlier works propose to directly optimize the discrete DAG structure through black-box optimization techniques such as reinforcement learning (Zoph & Le, 2016), evolutionary algorithms (Liu et al., 2017), and Bayesian optimization (Kandasamy et al., 2018), which are inherently less efficient. A more recent approach is to encode DAGs into some continuous space for searching, and various DAG encoders are developed. In general, these DAG encoding schemes fall into two categories: structure-aware encoding scheme (Ying et al., 2019; Wen et al., 2020; Shi et al., 2020) and computation-aware (performance-aware) encoding scheme (Zhang et al., 2019; Thost & Chen, 2021; Ning et al., 2020).

Due to the superior graph representation learning ability, graph neural networks (GNNs) have broadly achieved state-of-art performance on various graph learning tasks, such as node classification (Velickovic et al., 2018; Hamilton et al., 2017), graph classification (Xu et al., 2019; Zhang et al., 2018; Duvenaud et al., 2015), and link prediction (Zhang & Chen, 2018; Zhang et al., 2020). Basically, GNNs follow the message passing scheme (Gilmer et al., 2017) where each node aggregates node features from its one-hop neighborhood repeatedly to update its own feature, and this aggregation happens at all nodes simultaneously. However, Thost & Chen (2021) suggest that such a framework cannot exploit the inductive bias of the computation dependency defined by DAGs, thus failing to generate a smooth latent (DAG encoding) space beneficial to downstream optimization routines.

Hence, in order to model the dependency between nodes in DAGs, various GNNs specifically designed for encoding DAGs, such as D-VAE (Zhang et al., 2019) and DAGNN (Thost & Chen, 2021), are developed to inject the computation dependency between nodes into the representation learning process. Instead of updating node features simultaneously, these DAG encoders are constructed upon a gated recurrent unit (GRU) and will not update the representation of a node until all of

its predecessors are updated. One way to achieve this is to perform message passing sequentially following a topological ordering of the nodes. Such an asynchronous message passing scheme actually simulates how a real computation is performed along the DAG—the message passing order respects the computation dependency defined by the DAG, thus better exploiting the inductive bias. However, one key limitation of such DAG encoders is that the encoding process is inherently sequential, precluding processing all nodes in parallel. Although DAGNN proposes a topological batching trick to accelerate the training speed by partitioning nodes into disjoint batches where nodes within a batch can be processed in parallel, the time complexity is still lower-bounded by the longest path in the DAG and the fundamental constraint of the sequential computation nature still remains.

Numerous efforts have been made to reduce the sequential computation cost in sequence modeling literature (Cho et al., 2014; Wu et al., 2016). For instance, ConvS2S (Gehring et al., 2017) utilizes convolutional layers as building blocks to compute output representations at different positions, while Transformer (Vaswani et al., 2017) proposes to inject the position (order) information into the model through positional encoding, and then dependency between positions can be captured through the attention mechanism (Bahdanau et al., 2015; Gehring et al., 2017) in parallel instead of resorting to recurrent neural networks (RNNs). However, the success of these techniques relies on the inherent linear order of symbols in the input/output sequences that automatically characterizes the dependency between symbols. That is, the dependency between symbols is fully captured by their positions in the sequence. Such a condition is not satisfied by nodes in a DAG $G = (V, E)$ since each node can have **multiple predecessors** instead of only one like in plain sequences, and the dependency between nodes forms a (strong) partial order rather than a linear order. Thus, previous parallel sequence modeling methods would fail in the representation learning of DAGs.

In this paper, we propose a novel Parallelizable Attention-based Computation structure Encoder, PACE, to improve the computation efficiency over existing GRU-based DAG encoders. In order to borrow the power of Transformer for sequence modeling to DAG modeling problems, we need to design a *positional encoding* scheme specifically for DAGs which can fully capture the dependency between nodes in a DAG before applying the pairwise self-attention mechanism. To achieve this, we propose a GNN-based dag2seq framework which is proved to injectively map DAGs to sequences of node embeddings. This means, we are able to fully recover the DAG structure from these produced node embeddings, the same as the positional encoding in the original Transformer. After that, a Transformer encoder (with mask operation) is applied to the node embedding sequence to simultaneously learn representations for all nodes in the DAG through the self-attention mechanism. This way, PACE incorporates the relational inductive bias (Battaglia et al., 2018; Xu et al., 2020) carried by DAG structures into the encoding process, while eschewing the recurrence in previous works thus greatly improving the parallelization and encoding efficiency.

To demonstrate the superiority of the proposed PACE model, we evaluate PACE against current state-of-art DAG encoders as well as other general-purpose graph encoders for undirected graphs. Experimental results show that PACE not only outperforms competitive baselines by generating smooth latent spaces that capture the similarity between DAGs thus facilitating the downstream search subroutines, but also significantly boosts the training and inference speed through parallelization.

## 2 BACKGROUNDS

### 2.1 PARALLELIZABLE SEQUENCE MODELS

Encoding the complexities and nuances of sequences plays a central role in various machine learning tasks, including sentiment classification (Medhat et al., 2014), speech recognition (Abdel-Hamid et al., 2014), and other natural language processing (NLP) tasks (Khan et al., 2016). For many years, sequential models, such as recurrent neural networks (RNNs) (Medsker & Jain, 2001), were the primal way to solve the sequence encoding problem. These models are computationally expansive due to the sequential encoding process. Hence, many parallelizable sequence models are proposed, including Transformer (Vaswani et al., 2017), BERT (Devlin et al., 2019), etc. Our proposed PACE model is built upon the Transformer (encoder) architecture.

Transformer (Vaswani et al., 2017) is arguably the earliest translation model that solves the sequence-to-sequence task without using sequence-aligned RNNs or convolutional architectures. Transformer relates information from different positions in the sequence through the (masked) self-attention mech-

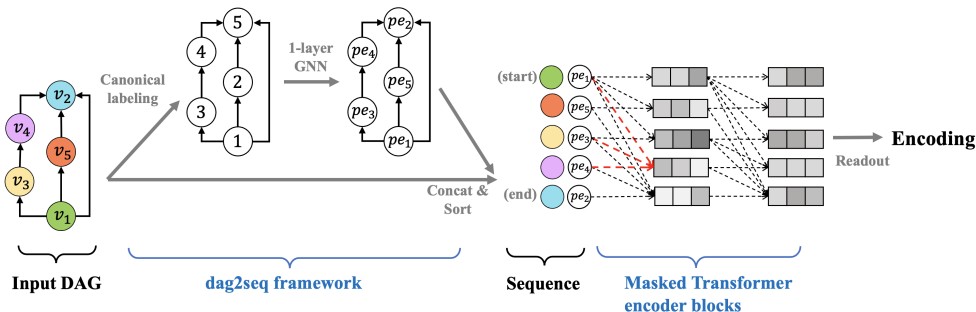

Figure 1: Illustration of PACE. The input DAG is first injectively represented as a sequence through the dag2seq framework, and then the sequence is fed into multiple stacked masked Transformer encoder blocks. The operations of nodes (i.e. node types) are visualized as colors, and nodes in the sequence is sorted according to the canonical label generated in the dag2seq framework.

anism to encode/decode the representation of items in the input/output sequence, and incorporates the order of the items into the encoding/decoding process through a positional encoding mechanism. Briefly, the positional encoding mechanism is an injective function $f_{pe} : \mathbb{N} \rightarrow \mathbb{R}^d$ that represents the positions (i.e. indices) of items in the sequence as $d$-dimensional vectors. Hence, it consistently outputs a unique encoding for each position in the sequence. The Transformer architecture is inherently parallelizable and could capture the long-term dependency with ease, thus is broadly applied to sequence modeling tasks in following works (Dai et al., 2019; Al-Rfou et al., 2019; Devlin et al., 2019; Lewis et al., 2020). Recently, Transformer has been widely used in image processing, and has achieved the state-of-art performance on various image learning tasks, including object detection (Carion et al., 2020) and image recognition (Dosovitskiy et al., 2020).

## 2.2 DAG ENCODING PROBLEM

A directed acyclic graph (DAG) $G$ is represented as a pair $(V, E)$ with $V = \{v_1, v_2, .., v_n\}$ denoting the set of nodes and $E \in V \times V$ denoting the set of directed edges. A DAG often carries a computation. We use $\mathcal{O}$ to denote an (computational) operation dictionary. For instance, the operation dictionary $\mathcal{O}$ for NAS-Bench-101 dataset contains five operations: "Input", "Output", "$3 \times 3$ convolution", "$1 \times 1$ convolution", and "$3 \times 3$ max-pool". Let $G = (V, E)$ be a DAG whose nodes represent operations in $\mathcal{O}$. Then the DAG $G$ represents a **computation structure** in which dependencies between operations are determined by directed edges in $E$. Hence, isomorphic DAGs define the same computation structure. Then the objective of the DAG encoding problem is to develop encoders that can generate embeddings to distinguish the computation structures defined by DAGs in the encoding space.

DAGs $G = (V, E)$ show a close relation with partial order. For any two nodes $v_i, v_j \in V$, let $\preceq$ be a binary relation such that $v_i \preceq v_j$ if and only if there is a directed path from $v_i$ to $v_j$, then the binary relation $\preceq$ defines a partial order on the node set $V$. Based on the partial order, sequential DAG encoders, such as D-VAE (Zhang et al., 2019) and DAGNN (Thost & Chen, 2021), use GRU (Cho et al., 2014) to recursively encode nodes in the input DAG, where a node is not encoded until all of its predecessors (those nodes with a $\preceq$ relation with it) are encoded. Because of the possibly very long dependency chains, these sequential DAG encoders inherently share the same limitations as RNNs, such as the slow training and inference speed and the difficulty to capture long-term dependencies.

To address these limitations, it is intuitive to generalize Transformer to the DAG encoding problem, since Transformer brings undoubtedly a huge improvement over the RNN-based sequence models. However, Figure 2 illustrates that it can be ambiguous how to represent DAGs as sequences due to the complex topological structure of DAGs. Let $o : V \rightarrow \mathcal{O}$ be the function mapping each node in $G$ to an operation in $\mathcal{O}$, and $f_{pe}$ be the original positional encoding function in Transformer. One intuitive way to linearize a DAG into a sequence is to sort its nodes with a topological order, which is also used in GRU-based DAG encoders. However, the topological orders of nodes in a DAG are often not unique. For instance, graph $G_1$ in Figure 2 has two different topological orders: $v_1, v_2, v_3, v_4, v_5$ and $v_2, v_1, v_3, v_4, v_5$, hence resulting in two different node sequences $Seq_1$ and $Seq_2$. Note that this does not hurt GRU-based DAG encoders as they are invariant to which topological order is used.

To avoid the ambiguity of topological order, another approach is to sort nodes according to a canonical order (i.e., node index in the canonical form of the DAG) as suggested by Niepert et al. (2016). For example, let $v_1, v_2, v_3, v_4, v_5$ be the nodes sorted by a canonical order. Then, we can linearize a DAG into a sequence $(o(v_1), f_{pe}(v_1)), (o(v_2), f_{pe}(v_2)), ...(o(v_n), f_{pe}(v_n))$, similar to how Transformer represents a sentence. Although canonical order guarantees the generated sequence is unique for the same DAG, different DAGs might have the same sequence. For instance, DAGs $G_1$ and $G_2$ in Figure 2 are not isomorphic, yet they will be represented as the same sequence $Seq_2$. This is because the positional encoding function

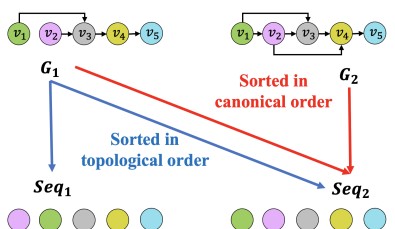

Figure 2: Illustration of the ambiguity when applying sequence models (such as Transformer) to the DAG encoding problem.

$f_{pe}$ only captures node positions in the sequence but fails to encode the node dependencies, causing a significant structural information loss.

In summary, it is not straightforward to generalize Transformer to DAGs. We need to design a smart linearization method for DAGs which guarantees that the same DAG is transformed to the same sequence, while different DAGs are always transformed to different sequences, in order to achieve a lossless transformation.

## 3 THE PACE MODEL

In this section, we describe the proposed **P**arallelizable **A**tention-based **C**omputation structure **E**ncoder (**PACE**). The key component of PACE is a **dag2seq** framework, which leverages graph canonization procedure and a graph neural network to transform DAGs into sequences while preserving the distinctiveness of different DAGs, so that the DAG encoding problem can be addressed efficiently by applying Transformer to the sequences.

### 3.1 THE DAG2SEQ FRAMEWORK

Here we describe the proposed dag2seq framework. Let $G = (V, E, o)$ be a labeled DAG (node labels are operations in $\mathcal{O}$), where $V = \{1, 2, ..., n\}$ is the finite set of nodes, $E$ is the set of directed edges, and $o : V \rightarrow \mathcal{O}$ is a function that associates to each node an operation in $\mathcal{O}$. We denote the *canonical form* of $G$ as $C(G) = (V^C, E^C, o^C)$, which assigns to each labeled graph $G$ an isomorphic labeled graph $C(G)$ that is a unique representation of its isomorphism class. That is, all labeled graphs isomorphic to $G$ will have the same canonical form $C(G)$. Since $G$ and $C(G)$ are isomorphic, there exists a bijection $\pi : V \rightarrow V^C$ (note that $V^C = \{1, 2, ..., n\}$) between the node sets such that $o^C(\pi(i)) = o(i)$ for all $i \in V$ and $(\pi(i), \pi(j)) \in E^C$ if and only if $(i, j) \in E$. The graph canonization process assigns a new index $\pi(i)$ to each node $i$. Based on the new indices $\pi(i)$, the proposed dag2seq computes the *positional encoding* of node $j \in V$ (denoted as $p_j$) as follows

$$a_j = Agg(\pi(i), (i, j) \in E) \tag{1}$$
$$p_j = Combine(\pi(j), a_j) \tag{2}$$

where functions *Agg* and *Combine* follow the same definition as Graph Isomorphism Network (GIN) (Xu et al., 2019). From Equations (1) and (2), we can see that dag2seq uses the canonical indices as node features and applies a one-layer injective GNN to obtain the positional encoding of each node. Note that $(i, j) \in E$ is equivalent to $(\pi(i), \pi(j)) \in E^C$. Hence, Equations (1) and (2) can also be interpreted as applying a one-layer injective GNN on the canonical graph $C(G)$ with the true node indices of $C(G)$ as node features. For notation convenience, we use $\pi^{-1} : V^C \rightarrow V$ to denote the inverse function of $\pi$. Then Theorem 3.1 describes how dag2seq generates sequences that uniquely represent DAGs.

**Theorem 3.1** *Let $G = (V, E, o)$ be a labeled DAG, and $p_1, p_2, ..., p_n$ be the positional encodings generated by dag2seq. If functions Agg and Combine are injective, then the sequence $(o(\pi^{-1}(1)), p_{\pi^{-1}(1)}), (o(\pi^{-1}(2)), p_{\pi^{-1}(2)}), ..., (o(\pi^{-1}(n)), p_{\pi^{-1}(n)})$ injectively encodes the computation structures defined by DAGs.*

We prove Theorem 3.1 in Appendix A. The significance of Theorem 3.1 is that it provides a way to injectively encode DAGs into sequences which fully preserves the node type information as well as structure information of the original DAGs. In other words, two labeled DAGs will be encoded into the same sequence if and only if they are isomorphic (their computation structures are the same). Then advanced parallelizable encoders (such as the Transformer encoder) for sequence modeling can be applied to the DAG encoding problem to improve the efficiency of the encoding process, thus facilitating the downstream DAG optimization problems. Intuitively, the canonical form provides a unified node indexing for isomorphic DAGs which initially may have different node orderings, while the one-layer injective GNN encodes the direct predecessors of each node into its positional encoding. Then it is not difficult to see that from the canonical indices of nodes and their direct predecessors as well as the node types we can fully recover the original DAG. The one-layer GNN is parallelizable, in contrast to the sequential GNNs used in previous works (Zhang et al., 2019; Thost & Chen, 2021).

It is also worth discussing the complexity of graph canonization. The graph canonization problem is theoretically at least as computationally hard as the graph isomorphism problem, which is in NP but not known to be solvable in polynomial time nor to be NP-complete. However, getting the canonical form of graphs is not too difficult in practice, thanks to the well-known graph canonization tools such as Nauty (McKay & Piperno, 2014). Empirically, such tools usually return the canonical form of a reasonable-sized graph in seconds. Theoretically, Nauty has an average time complexity of $O(n)$, and polynomial-time graph canonization algorithms also exist for graphs of bounded degrees. We also found in our experiments that graph canonization adds a negligible overhead.

## 3.2    THE TRANSFORMER IN PACE

With the dag2seq framework to injectively map DAGs to sequences, we next develop the attention-based parallelizable encoder which consists of $K$ stacked Transformer (encoder) blocks.

For each item $(o(\pi^{-1}(i)), p_{\pi^{-1}(i)})$ in the output sequence of dag2seq, $o(\pi^{-1}(i))$ provides the operation information, while the positional encoding $p_{\pi^{-1}(i)}$ contains the structural information. Hence, PACE concatenates the trainable embedding of operation $o(\pi^{-1}(i))$ and the positional encoding $p_{\pi^{-1}(i)}$ as the embedding $e_i$ of item $i$ in the sequence.

$$e_i = Concat(Emb(o(\pi^{-1}(i))), p_{\pi^{-1}(i)}) \tag{3}$$

Then the sequence $e_1, e_2, ..., e_n$ is fed into the first Transformer encoder block. Each transformer encoder block performs the multi-head self-attention mechanism (Vaswani et al., 2017) to update the embedding of each item in the sequence. We provide details of the multi-head self-attention mechanism in Appendix C.

In the original transformer encoder blocks, the attention operations are not masked. In other words, for any two items $i$ and $j$ in the sequence, the embedding of item $i$ will be used to update the embedding of item $j$. Such unmasked operation is reasonable as the positional encodings $\{p_{\pi^{-1}(i)}, \forall i\}$ have already encoded the partial order between nodes. However, due to the complex dependencies a DAG may encode, fully relying on the positional encodings to capture such dependencies might not be enough. Therefore, we introduce a masked attention operation that helps better capture the dependency between nodes in practice. The masked attention operation can be specified through a binary mask matrix $M$ as follows.

$$M_{i,j} = \begin{cases} \textit{False,} & \textit{if there exists a path from } i \textit{ to } j \textit{ in } C(G) \\ \textit{True,} & \textit{otherwise} \end{cases} \tag{4}$$

In this mask matrix $M$, element $M_{i,j} = \textit{True}$ indicates that the effect of item $i$ in updating the embedding of item $j$ will be masked out. When $M_{i,j} = \textit{True}$, there is no (directed) path from node $i$ to node $j$ in the canonical form $C(G)$. In other words, we only allow $j$'s predecessors in $C(G)$ (or equivalently, $\pi^{-1}(j)$'s predecessors in $G$) to participate in the updating of $j$'s embedding. Such a masking operation has two benefits: 1) the structure information of the DAG is strengthened in the masked self-attention, and 2) the partial order between operations in the computation structure is exploited, which aligns with the logic of a real computation in the sense that the operation at some node does not depend on its successor operations. We also empirically verify the effectiveness of the masked attention operation in the ablation study. The mask matrix $M$ can be efficiently computed through the DFS algorithm or the Floyd algorithm, which we describe in Appendix B.

### 3.3 Training Methodology

We design two ways to train the PACE model. One is by training a variational autoencoder (VAE) for DAGs that is able to encode and decode/generate DAGs into and from a latent space, like D-VAE and DAGNN. On the other hand, since the proposed dag2seq transforms the DAG encoding problem to the sequence encoding problem, pre-training techniques in NLP are also suitable for training PACE.

Without loss of generality, we assume that there is a single output node in each DAG that has no successor. If not, we can add a virtual output node and add directed edges from all nodes whose out-degree is $0$ to the output node. When PACE is trained in a VAE architecture, we use a common trick in standard Transformers by assuming that there are at most $N$ nodes in the input DAG. If a DAG has $n < N$ nodes, we pad $N - n$ end symbols to the end of the sequence generated by dag2seq. Then PACE readouts the DAG encoding by concatenating the learned embeddings of the $N$ symbols. When PACE is trained in a pre-training architecture, similar to the sentiment classification task in BERT, PACE takes the learned embedding of the output node as the DAG encoding.

**Training PACE in a VAE architecture.** In the PACE-VAE architecture, we take PACE as the encoder, and connect the output of PACE with two fully connected (FC) layers to predict the mean and variance of the approximated posterior distribution in the evidence lower bound (ELBO) (Kingma & Welling, 2013). The decoder of PACE-VAE consists of $K = 3$ standard Transformer decoder blocks of dimension $d_k$. Given the latent vector $z$ to decode, the decoder uses an FC layer to reconstruct a vector of dimension $N \times d_k$, and the vector is then reshaped to a matrix $Z$ of shape $(N, d_k)$, which plays the same role as the "memory" matrix in a standard Transformer in NLP. Hence, the encoder-decoder attention layer in each Transformer decoder block takes the decoded matrix $Z$ as the "Key" matrix and "Value" matrix, and uses the output from the previous self-attention layer as the "Query" matrix. Similar to the standard Transformer, during the generation, the decoder of PACE-VAE sequentially predicts nodes in $G$ according to the learnt canonical order, and this process is ended until a special symbol is predicted indicating the decoding process is completed. For each generated node $i$ in the decoding process, we do a softmax to select the operation of the node, and use a binary classifier to predict the existence of an edge between node $i$ and any node $j < i$. We describe more details of PACE-VAE and its parallelizable training framework in Appendix D.

**Training PACE in a pre-training architecture.** After converting DAGs to sequences by the proposed dag2seq framework, PACE is essentially a sequence modeling encoder. Yan et al. (2020) validates that the pre-training architecture in NLP that generates embeddings without using accuracies can better preserve the local structural relationship in the latent space. As such, PACE can also take the masked language modeling (MLM) (Devlin et al., 2019; Yan et al., 2021) objective for pre-training to capture the locality information of the computation structure defined by DAGs. For each input DAG, we randomly select 20% nodes for masking and prediction, where 80% of them are replaced with the $[MASK]$ token and the remaining nodes are unchanged. The output embeddings are used to predict the original node operations $o(\pi^{-1}(i))$, and we train PACE by minimizing the cross-entropy loss of the predicted node operations and the true node operations.

## 4 Comparison to Related Works

Despite the great success of general-purpose GNNs in encoding undirected graphs, there have been relatively fewer GNN works proposed to capture the inductive bias encoded by the dependency between nodes in DAGs. S-VAE (Bowman et al., 2016) takes as input the sequence of node strings which consist of the node type as well as the adjacency vector of each node, and then applies a GRU-based RNN to the topologically sorted node sequence to encode a DAG. Following this idea, D-VAE (Zhang et al., 2019) proposes a DAG-style message passing framework that sequentially updates the DAG encoding following a topological order of nodes. For each node in a DAG, D-VAE takes a gated summation aggregator to combine information from all its direct predeccessors, and then a GRU is used to update the node embedding based on the aggregated information and node type. Similar to the encoder of D-VAE, Thost & Chen (2021) propose DAGNN which also sequentially aggregates node features. It differs from D-VAE in the sense that the aggregator is constructed through the attention mechanism and it comes with a layer notion in the DAG encoding process.

The proposed PACE model is closely related to these DAG encoders. However, there is a noticeable difference. Previous DAG encoders (i.e., S-VAE, D-VAE, DAGNN) share a major limitation that

the sequential nature of the encoder precludes their parallelizability. In constrast, PACE adopts a dag2seq framework that encodes the dependencies between nodes in the positional embeddings, and then applies a transformer to the node sequence to encode a DAG parallelly.

# 5 EXPERIMENTS

In this section, we conduct experiments on popular DAG encoding datasets to validate the effectiveness and efficiency of the proposed PACE model against state-of-art DAG encoders and other general-purpose (undirected) graph encoders.

## 5.1 DATASETS AND METRICS

**NA and BN.** The NA dataset consists of approximately 19K neural architectures generated by the software `ENAS` (Pham et al., 2018). Each architecture has its pre-computed weight-sharing (WS) accuracy on CIFAR-10 (Krizhevsky et al., 2009), and includes 8 nodes with the first node as the input node and the last node as the output node. The BN dataset consists of 200K Bayesian networks randomly generated by the `bnlearn` package (Scutari, 2010). Each Bayesian network has 8 nodes and is associated with a Bayesian Information Criterion (BIC) score that measures the architecture performance on the Asia (Lauritzen & Spiegelhalter, 1988) dataset.

On NA and BN, following the experimental setting used in (Zhang et al., 2019; Thost & Chen, 2021), the PACE model is evaluated under a VAE architecture, and we take $90\%$ NA/BN data as the training set and hold out the rest for testing. To make a fair comparison, we evaluate the quality of DAG encoders by measuring the prediction performance and the downstream search performance as suggested by Zhang et al. (2019). Briefly, a sparse Gaussian Process (SGP) regression model (Snelson & Ghahramani, 2005) is trained to predict the DAG performance from its encoding, and we use rooted mean square error (RMSE) and Pearson correlation (Pearson'r) as metrics to evaluate the prediction performance. When evaluating the downstream search performance, we perform Bayesian optimization (BO) in the DAG encoding space based on the SGP regression model, and compare the performance of the best searched architecture.

**NAS101 and NAS301.** NAS101 (NAS-Bench-101) and NAS301 (NAS-Bench-301) are two well-known neural architecture search (NAS) benchmark datasets. NAS101 (Ying et al., 2019) consists of approximately 420K neural architectures with pre-computed validation and test accuracies on CIFAR-10, where each architecture has up to 7 nodes and 9 edges with the first node as the input and the last node as output node. NAS301 (Siems et al., 2020) is a surrogate benchmark for DARTS (Liu et al., 2018b). Following Liu et al. (2018a); Yan et al. (2021), we randomly sample 1M neural architectures, where each architecture contains at most 15 nodes.

On NAS101 and NAS301, the PACE model is trained with the pre-training architecture using the MLM objective. Since better prediction performance of DAG encoders always facilitates the downstream search performance, we implement two popular BO-based downstream search methods, DNGO (Snoek et al., 2015) and DNGO-LS (Yan et al., 2021), and compare the downstream search performance of different encoders. For NAS101, following the original work of Ying et al. (2019), we take regret as the metric to evaluate the downstream performance, where the regret is the difference between the test accuracy of the (offline) optimal neural architecture and the test accuracy of the best searched neural architecture (after 20 rounds). For NAS301, since we do not have an oracle for the optimal neural architecture, we use the test accuracy of the best searched neural architecture, instead.

## 5.2 BASELINES AND MODEL CONFIGURATION

On NA and BN, we compare PACE to all baselines used in DAGNN: D-VAE, S-VAE (Bowman et al., 2016), GCN (Zhang et al., 2019), GraphRNN (You et al., 2018), DeepGMG (Li et al., 2018). On NAS101 and NAS301, we compare PACE with three undirected graph encoders: GIN (Xu et al., 2019), GAT (Velickovic et al., 2018), GCN (Kipf & Welling, 2016), and current state-of-art DAG encoders: S-VAE, D-VAE, DAGNN.

The PACE model always uses 3 Transformer encoder blocks to boost the training and inference speed. We use an embedding layer of dimension 64 to map node types to node type embeddings. The output dimension of the 1-layer GNN in our dag2seq framework is also 64. On NA and BN, we concatenate

Table 1: Predictive performance on NA and BN.

| Evaluation Metric | NA | | BN | |
|---|---|---|---|---|
| | RMSE ↓ | Pearson's r ↑ | RMSE ↓ | Pearson's r ↑ |
| **PACE** | **0.254 ± 0.002** | **0.964 ± 0.001** | **0.115 ± 0.004** | **0.994 ± 0.001** |
| DAGNN | 0.264 ± 0.004 | 0.964 ± 0.001 | 0.122 ± 0.004 | 0.991 ± 0.000 |
| D-VAE | 0.384 ± 0.002 | 0.920 ± 0.001 | 0.281 ± 0.004 | 0.964 ± 0.001 |
| S-VAE | 0.478 ± 0.002 | 0.873 ± 0.001 | 0.499 ± 0.006 | 0.873 ± 0.002 |
| GraphRNN | 0.726 ± 0.002 | 0.669 ± 0.001 | 0.779 ± 0.007 | 0.634 ± 0.001 |
| DeepGMG | 0.478 ± 0.002 | 0.873 ± 0.001 | 0.843 ± 0.007 | 0.555 ± 0.003 |
| GCN | 0.832 ± 0.001 | 0.527 ± 0.001 | 0.599 ± 0.006 | 0.809 ± 0.002 |

Table 2: Downstream search performance on NA and BN.

| Model | PACE | DAGNN | D-VAE | S-VAE |
|---|---|---|---|---|
| (NA) Test accuracy ↑ | **95.08** | 95.06 | 94.80 | 92.79 |
| (BN) Optimized BIC score ↑ | **-11107.29** | **-11107.29** | −11125.75 | −11125.77 |

the positional encoding and node type embedding as the node features fed into the first Transformer encoder block. On NAS101 and NAS301, we use the summation of positional encoding and node type embedding, instead. All the experiments are done on NVIDIA Tesla P100 12GB GPUs.

## 5.3 EXPERIMENTAL RESULTS

**NA and BN.** According to Table 1, PACE achieves the smallest RMSE and the largest Pearson's r on both NA and BN, indicating that PACE generates the smoothest searching space that captures the locality of DAGs. Furthermore, we notice that the improvement is more significant on NA. One possible reason is that DAGs in NA always have a Hamiltonian path which introduces relatively long-term dependency between nodes. Since the attention mechanism in PACE could capture the long-term dependency better compared to RNN-based DAG encoders, PACE can preserve the similarity of DAGs better in the learned DAG embeddings. Furthermore, as Table 2 shows, PACE detects architectures with the best performance on both NA and BN. Such an observation indicates that the smooth DAG embedding space can facilitate the downstream searching subroutines. In addition, we also visualize the optimal detected architectures in Appendix E. Finally, in analogy to D-VAE, we also compare the reconstruction accuracy and the generation performance (i.e., the proportions of valid/ unique/ novel architectures in the generated DAGs) of the tested DAG autoencoders. It turns out that PACE still achieves the best performance, and the results are presented in Appendix F.

**NAS101 and NAS301.** Table 3 presents the results on NAS101 and NAS301. PACE significantly outperforms undirected graph encoders and other DAG encoders on each dataset and each searching method. Similar to PACE, DAGNN and GAT also use the attention mechanism to model the dependencies (relations) between nodes. However, the attention mechanism in these encoders is only applied to nodes and their direct predecessors (DAGNN) or adjacent nodes (GAT), hence making it hard to capture the long-term dependency between nodes. On the contrary, PACE allows all predecessor nodes to participate in the attention mechanism while encoding the neighbors through positional encoding, which enables PACE to learn long-term dependencies as well as short-term dependencies with ease, thus capturing the similarity of computation structure defined by DAGs more efficiently and more effectively.

## 5.4 COMPUTATIONAL COST

A key advantage of PACE is the parallelizable DAG encoding process, so we compare the computational cost of PACE to GRU-based DAG encoders (D-VAE and DAGNN). We use a single GPU for each experiment, and Figure 3 shows our results. On datasets NA and BN, PACE, D-VAE and DAGNN are all trained with the VAE architecture. As all models are trained in the same manner, both the training and inference time can reflect the DAG encoding speed. Hence we compare the average training time per epoch and the total required training epochs to reach optimal performance to evaluate the computational cost. Figure 3 (a) and (b) show that PACE requires significantly

Table 3: Experimental results on NAS101 and NAS301.

| Search Method | NAS101 (Regret) | | NAS301 (Acc) | |
|---|---|---|---|---|
| | DNGO (%) ↓ | DNGO-LS (%) ↓ | DNGO (%) ↑ | DNGO-LS (%) ↑ |
| PACE | **0.391 ± 0.241** | **0.278 ± 0.178** | **94.507 ± 0.165** | **94.547 ± 0.145** |
| DAGNN | 0.445 ± 0.224 | 0.448 ± 0.127 | 94.445 ± 0.219 | 94.433 ± 0.156 |
| D-VAE | 0.439 ± 0.203 | 0.430 ± 0.222 | 94.453 ± 0.148 | 94.428 ± 0.131 |
| S-VAE | 0.458 ± 0.175 | 0.451 ± 0.225 | 94.332 ± 0.183 | 94.371 ± 0.203 |
| GIN | 0.593 ± 0.177 | 0.518 ± 0.201 | 94.451 ± 0.224 | 94.411 ± 0.198 |
| GAT | 0.597 ± 0.269 | 0.509 ± 0.187 | 94.430 ± 0.171 | 94.421 ± 0.202 |
| GCN | 0.627 ± 0.161 | 0.538 ± 0.233 | 94.448 ± 0.149 | 94.404 ± 0.160 |

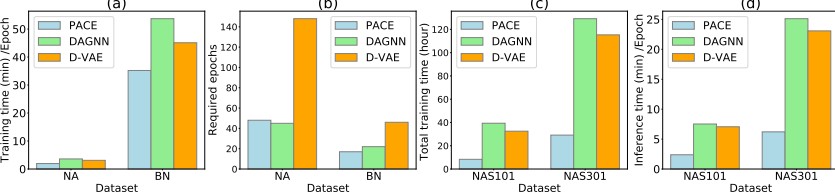

Figure 3: Computational cost.

less training time per epoch than D-VAE and DAGNN, while reaching optimal performance with fewer epochs on average. On NAS101 and NAS301, as PACE uses a different training method (i.e., pre-training), we compare the overall training time and the average inference time per epoch to make a fair comparison. Figure 3 (c) and (d) indicate that PACE takes about $\frac{1}{4}$ average training time and roughly $\frac{1}{4}$ average inference time compared to GRU-based DAG encoders (D-VAE, DAGNN). Overall, PACE significantly reduces the computational cost against previous DAG encoders.

## 5.5 ABLATION STUDY

In the ablation study, we demonstrate the effectiveness of our proposed dag2seq (positional encoding) framework and the attention mask in PACE. From Table 4, we have the following observations: 1) In general, PACE trained with attention mask outperforms the one without attention mask, indicating that the attention mask helps better capture the inductive bias of DAGs. Nevertheless, even without attention mask, PACE still performs relatively well because the dag2seq framework also captures the node dependencies. 2) We also find that the dag2seq framework is vital for the performance of PACE. Without dag2seq, PACE (without mask) almost completely fails at the NA dataset. This verifies the importance of dag2seq for solving the ambiguity issue of DAG encoding illustrated in Figure 2.

Table 4: Ablation study.

| Model configuration | NA | | BN | | NAS101 (Regret) | | NAS301 (Acc) | |
|---|---|---|---|---|---|---|---|---|
| | RMSE ↓ | Pearson's r ↑ | RMSE ↓ | Pearson's r ↑ | DNGO (%) ↓ | DNGO-LS (%) | DNGO (%) ↑ | DNGO-LS (%) ↑ |
| Model1: dag2seq & Mask | **0.254** | 0.964 | **0.115** | **0.9942** | **0.391** | **0.278** | **94.507** | **94.547** |
| Model2: dag2seq & No Mask | 0.255 | **0.967** | 0.119 | 0.9941 | 0.479 | 0.368 | 94.501 | 94.505 |
| Model3: No dag2seq & No mask | 0.981 | 0.001 | 0.368 | 0.9318 | 0.600 | 0.498 | 94.467 | 94.401 |

## 6 CONCLUSION

In this paper, we have proposed PACE, a novel DAG encoder based on Transformer. Unlike traditional RNN-based DAG encoders which sequentially encode DAG nodes, PACE is fully parallelizable, thus having a much better encoding speed. PACE incorporates the strong inductive bias through a node-dependency-aware positional encoding framework, dag2seq, and a masked self-attention mechanism. Experiments demonstrate that PACE not only generates smooth latent (DAG encoding) space beneficial to the downstream search, but also boosts the training and inference time significantly.

## 7 REPRODUCIBILITY STATEMENT

The main theoretical contribution of our paper comes from Theorem 3.1. Clear explanations and complexity analysis is provided in Section 3.1, while the complete proof of the Theorem is available in Appendix A. Furthermore, our proposed DAG encoder, PACE, incorporates the mask operation in the multi-head self-attention mechanism, hence we describe algorithms to get the mask matrix in Appendix B, and mathematically formulate the masked multi-head self-attention mechanism in Appendix C. In addition, as PACE can be trained in a novel VAE architecture that supports parallelizable training, we thoroughly explain the VAE architecture in Appendix D. All datasets (i.e. NA, BN, NAS101, NAS301) used in our experiments are public and we provide clear explanations of these datasets in Section 5.1. Our source code, which includes details of necessary data preprocessing, is provided in the supplementary materials, and we will make it public on Github in the future.

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

# A   PROOF OF THEOREM 3.1

Let $G_1 = (V_1, E_1, o_1)$ and $G_2 = (V_2, E_2, o_2)$ be two labelled graphs, then $G_1$ and $G_2$ are isomorphic (i.e.$G_1$ and $G_2$ represent the same computation structure.) if and only if their canonical form are identical, i.e. $C(G_1) = C(G_2)$. Note that it means equality between the canonical forms, not isomorphism. Let $C(G_1) = (V_1^C, E_1^C, o_1^C)$ and $C(G_2) = (V_2^C, E_2^C, o_2^C)$, then there exists bijections $\pi_1 : V_1 \to V_1^C$ and $\pi_2 : V_2 \to V_2^C$, and we use $\pi_1^{-1} : V_1^C \to V_1$ and $\pi_2^{-1} : V_2^C \to V_2$ to denote their inverse functions. Hence, we have $C(G_1) = C(G_2)$ if and only if (1) $o_1^C(i) = o_2^C(i)$ for $\forall i$; and (2) $(i, j) \in E_1^C \Leftrightarrow (i, j) \in E_2^C$ for $\forall i, j$.

Next, we will prove Theorem 3.1 by equivelantly showing that the sequence $(o(\pi^{-1}(1)), p_{\pi^{-1}(1)}), (o(\pi^{-1}(2)), p_{\pi^{-1}(2)}), ..., (o(\pi^{-1}(n)), p_{\pi^{-1}(n)})$ can guarantee the distinctness of canonical forms $C(G)$. For the notation convenience, let function $f(\pi(j), \{\pi(i), (i, j) \in E\}) = Combine(\pi(j), Agg(\pi(i), (i, j) \in E))$ be the composition of function Agg and Combine, then it is injective if and only if both Agg and Combine are injective. Forthermore, we use $Seq_1$ to denote the sequence $(o_1(\pi_1^{-1}(1)), p_{\pi_1^{-1}(1)}), (o_1(\pi_1^{-1}(2)), p_{\pi_1^{-1}(2)}), ..., (o_1(\pi_1^{-1}(n)), p_{\pi_1^{-1}(n)})$, and $Seq_2$ to denote the sequence $(oo_2(\pi_2^{-1}(1)), p_{\pi_2^{-1}(1)}), (o_2(\pi_2^{-1}(2)), p_{\pi_2^{-1}(2)}), ..., (o_2(\pi_2^{-1}(n)), p_{\pi_2^{-1}(n)})$.

So far, we know $C(G_1) \neq C(G_2) \Leftrightarrow$ there (1) exist $i$ such that $o_1^C(i) \neq o_2^C(i)$, or (2) exist $i, j$ such that $(i, j) \in E_1^C$ but $(i, j) \notin E_2^C$ (equivalently, $(i, j) \notin E_1^C$ but $(i, j) \in E_2^C$).

Now, let's prove $C(G_1) \neq C(G_2) \Rightarrow Seq_1 \neq Seq_2$.

- **(1) For the first case**, $\pi_1, \pi_2$ are the bijections that map $G_1$ and $G_2$ to their canonical forms, then we have:

$$o_1(\pi_1^{-1}(i)) = o_1^C(\pi_1(\pi_1^{-1}(i)))$$
$$= o_1^C(i)$$
$$o_2(\pi_2^{-1}(i)) = o_2^C(\pi_2(\pi_2^{-1}(i)))$$
$$= o_2^C(i)$$

  Since $o_1^C(i) \neq o_2^C(i)$, then we get $o_1(\pi_1^{-1}(i)) \neq o_2(\pi_2^{-1}(i))$. indicating that $Seq_1 \neq Seq_2$.
- **(2) For the second case**, according to the definition of canonical form, we know that $(\pi_1^{-1}(i), \pi_1^{-1}(j)) \in E_1 \Leftrightarrow (i, j) \in E_1^C$ (similarly,$(\pi_2^{-1}(i), \pi_2^{-1}(j)) \in E_2 \Leftrightarrow (i, j) \in E_2^C$). As such, we get:

$$p_{\pi_1^{-1}(j)} = f(\pi_1(\pi_1^{-1}(j)), \{\pi_1(\pi_1^{-1}(s)), (\pi_1^{-1}(s), \pi_1^{-1}(j)) \in E_1\})$$
$$= f(j, \{s, (s, j) \in E_1^C\})$$
$$p_{\pi_2^{-1}(j)} = f(\pi_2(\pi_2^{-1}(j)), \{\pi_2(\pi_2^{-1}(s)), (\pi_2^{-1}(s), \pi_2^{-1}(j)) \in E_1\})$$
$$= f(j, \{s, (s, j) \in E_2^C\})$$

  Then, since $(i, j) \in E_1^C$ but $(i, j) \notin E_2^C$, we have $\{s, (s, j) \in E_1^C\} \neq \{s, (s, j) \in E_2^C\}$. Since function $f$ is injective, then we have $p_{\pi_1^{-1}(j)} \neq p_{\pi_2^{-1}(j)}$. Hence, $Seq_1 \neq Seq_2$

In the end, let's prove the other direction, i.e. $Seq_1 \neq Seq_2 \Rightarrow C(G_1) \neq C(G_2)$. When $Seq_1 \neq Seq_2$, there must (1) exist $i$ such that $o_1(\pi_1^{-1}(i)) \neq o_2(\pi_2^{-1}(i))$, or (2) exist $j$ such that $p_{\pi_1^{-1}(j)} \neq p_{\pi_2^{-1}(j)}$.

- **(1) For the first case**, as previous analysis, we have

$$o_1^C(i) = o_1(\pi_1^{-1}(i))$$
$$o_2^C(i) = o_2(\pi_2^{-1}(i))$$

  Hence, we can get $o_1^C(i) \neq o_2^C(i)$, which indicates $C(G_1) \neq C(G_2)$.
- **(2) For the second case**, according to previous analysis, we know that:

$$p_{\pi_1^{-1}(j)} = f(j, \{s, (s, j) \in E_1^C\})$$
$$p_{\pi_2^{-1}(j)} = f(j, \{s, (s, j) \in E_2^C\})$$

Since $f$ is injective, $p_{\pi_1^{-1}(j)} \neq p_{\pi_2^{-1}(j)}$ implies that $\{s, (s,j) \in E_1^C\} \neq \{s, (s,j) \in E_2^C\}$, which indicates that there exists $i$ such that $(i,j) \in E_1^C$ but $(i,j) \notin E_2^C$ (or $(i,j) \notin E_1^C$ but $(i,j) \in E_2^C$). Henceforth, we get $C(G_1) \neq C(G_2)$.

## B  MASK MATRIX

Here we provide two potential ways to get the mask matrix in PACE. Following the same notation as the main paper, we use $C(G) = (V^C, E^C, o^C)$ to denote the canonical form of the input DAG $G$.

**DFS Algorithm**  This algorithm takes the canonical form $C(G)$ as input and performs the DFS algorithm on the graph to explore all the nodes of the graph. Before we start the deep first search, we traverse all edges in $E^C$ to find the direct-successors of each node $i$, and then put them in a set $S(i)$. Then, for each node $i$, we perform the DFS to get a dependent set $D(i$, and we have $M_{i,j} = $ *False* if and only if $j \in D(i)$.

---

**Algorithm 1** DFS Algorithm

---
1: Initialization: $D(i) = \{\}$; *Visited* = [*False for* $i \in V^C$]; a source (start) node $i$, $T = [i]$ (T is a stack).
2: *Visited*$[i] = $ *True*
3: **while** $|T| > 0$ **do**
4:     $j = T[-1]$
5:     delete $j$ from $T$
6:     **for** $k$ in $S(j)$ **do**
7:         **if** *Visited*$[k] = $ *Flase* **then**
8:             put $k$ in $D(i)$
9:             *Visited*$[k] = $ *True*
10:            put $k$ in $T$
11:         **end if**
12:     **end for**
13: **end while**

---

**Floyd Algorithm** The Floyd algorithm is originally proposed to for finding shortest paths in directed weighted graphs. Here, we initialize the edge weights to be 1, and implement the Floyd algorithm to find the distance $dist(i,j)$ (i.e. length of the shortest directed path) between each node pair $i,j$ in $C(G)$. Then we have $M_{i,j} = $ *False* if and only if $dist(i,j) > 0$.

---

**Algorithm 2** Floyd Algorithm

---
    Initialization: $dist(i,j) = 1$ if $(i,j) \in E^C$ else 0
2: **for** i $\in V^C$ **do**
    **for** j $\in V^C$ **do**
4:         **for** k $\in V^C$ **do**
            **if** $dist(j,k) > dist(j,i) + dist(i,k)$ **then**
6:                 $dist(j,k) = dist(j,i) + dist(i,k)$
            **end if**
8:         **end for**
    **end for**
10: **end for**

---

## C  MULTI-HEAD SELF-ATTENTION MECHANISM

Here we introduce the multi-head (masked) self-attention attention mechanism in the Transformer encoder blocks of PACE. For notation convenience, we use $H_k$ to denote the output representation of the $k$th Transformer encoder block, and use $H_0$ to denote the input (i.e. the representation of the sequence generated by dag2seq) to the first Transformer encoder block. Furthermore, we denote the

number of heads in the self-attention mechanism as $h$, and the embedding dimension (of each item in the sequence) as $d$. Then the Transformer encoder blocks update representation $H_k$ as following.

$$H_k^j = softmax(\frac{Q_k^j(K_k^j)^T}{d})V_k^j \quad for \ j = 1, 2, ...h \tag{5}$$

$$H_{k+1} = \textit{feed-forward}(\|_{j=1}^h H_k^j) \tag{6}$$

where $Q_k^j = H_k W_{k,q}^j$, $K_k^j = H_k W_{k,k}^j$, $V_k^j = H_k W_{k,v}^j$ are the query matrix, key matrix, value matrix, respectively (i.e. $W_{k,q}^j, W_{k,k}^j, W_{k,v}^j$ are trainable parameter matrices); $\|$ represents the concatenation operation; Feed-forward is a one-layer MLP. When we introduce the mask operation into the Transformer encoder block. let $M$ be the mask matrix from the Floyd algorithm or the BFS algorithm, then we use following equation to replace equation 5 in the Transformer encoder block.

$$H_k^j = softmax(\frac{Q_k^j(K_k^j)^T + -\infty * M}{d})V_k^j \quad for \ j = 1, 2, ...h \tag{7}$$

## D  MORE DETAILS ABOUT PACE IN THE VAE ARCHITECTURE

In the section, we describe the decoder of PACE-VAE. Figure 4 illustrates the overall architecture. In the main paper, we have introduced how PACE maps input DAGs to the latent space, here we focus on the decoder of PACE-VAE.

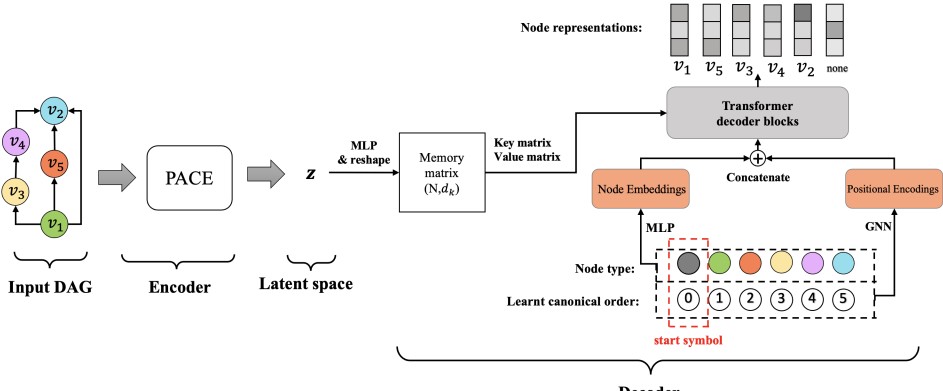

Figure 4: The illustration of PACE in the VAE architecture (PACE-VAE)

Similar to PACE, the decoder is constructed upon the Transformer decoder block. Each Transformer decoder block consists of a masked multi-head self-attention layer (i.e. Euqtion 7), a multi-head attention layer (i.e. Equation 5 except that the key matrix and value matrix are computed from points $z$ in the latent space), and a feed-forward layer (i.e. Equation 6). The decoder takes a MLP as the embedding layer to generate node type embeddings as PACE. In analogous to the dag2seq framework in PACE, the decoder also uses a GNN to generate the positional encoding based on the learnt canonical order of nodes. Then the node embeddings and positional encodings are concatenated and then fed into multiple consecutive Transformer decoder blocks to predict the node representations, which is used to predict the node types and the existence of edges. In analogous to the standard Transformer decoder, the decoder performs the shift right trick (i.e. the $i$th outputed node representation corresponds to the $i+1$th node in the sequence) and adds a start symbol node (i.e. the black node in Figure 4) at the beginning of the node sequence. Specifically, the canonical label of the start symbol node is different from any possible canonical label in the dag2seq framework to distinguish it's position. For instance, DAG in the searching space contains at most $N$ nodes, then the canonical order of the start symbol node can be 0 or $N+1$. Let $o_i$ denote the output representation of node $i$ in the sequence, then it is used to predict the type of node $i+1$ in the sequence through a MLP. Similarly, for any $j < i$, we use another MLP, which takes the concatenation of $o_j$ and $o_i$ as input, to predict the existence of an directed edge from node $j+1$ to node $i+1$ in the sequence. Note that the canonical order can be generated from the topological sort by breaking ties using canonicalization

tools, such as Nauty. Thus, for each node $i$ in the sequence, any dependent node $j$ of this node must be arranged in a prior position in the sequence (i.e. $j < i$). In the end, based on these predictions (node representations), we can perform the teacher forcing to train the VAE.

## E VISUALIZATION OF DETECTED OPTIMAL ARCHITECTURES ON NA AND BN

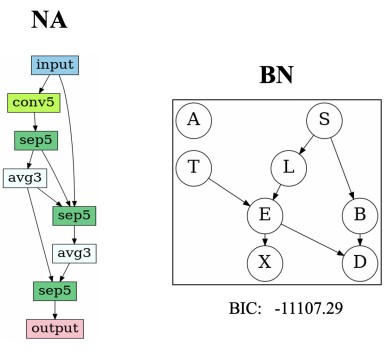

Figure 5: Best architectures on NA and BN detected by PACE.

In this section, we visualize the optimal architectures detected by Bayesian optimization (over the latent DAG encoding space generated by PACE) on datasets NA and BN. Figure 5 illustrates our results. On dataset BN, we find that the detected optimal Bayesian network structure is almost the same as the ground truth (Figure 2 of (Lauritzen & Spiegelhalter, 1988)). In the ground truth, there is another directed edge from node A (visit to Asia ?) to node T (Tuberculosis).

## F RECONSTRUCTION ACCURACY AND GENERATION PERFORMANCE COMPARISON

Table 5: Recon. accuracy, valid prior, uniqueness, novelty and overall (ave) performance %

| Methods | NA | | | | | BN | | | | |
|---|---|---|---|---|---|---|---|---|---|---|
| | Accuracy ↑ | Valid ↑ | Unique ↑ | Novel ↑ | **Overall ↑** | Accuracy ↑ | Valid ↑ | Unique ↑ | Novel ↑ | **Overall ↑** |
| PACE | 99.97 | 98.16 | 57.77 | 100.00 | **88.98** | 99.99 | 99.96 | 45.10 | 98.50 | **85.88** |
| DAGNN | 99.97 | 99.98 | 37.36 | 100.00 | 84.33 | 99.96 | 99.89 | 37.61 | 98.16 | 83.91 |
| D-VAE | 99.96 | 100.00 | 37.26 | 100.00 | 84.31 | 99.94 | 98.84 | 38.98 | 98.01 | 83.94 |
| S-VAE | 99.98 | 100.00 | 37.03 | 99.99 | 84.25 | 99.99 | 100.00 | 35.51 | 99.70 | 83.80 |
| GraphRNN | 99.85 | 99.84 | 29.77 | 100.00 | 82.37 | 96.71 | 100.00 | 27.30 | 98.57 | 80.65 |
| GCN | 5.42 | 99.37 | 41.48 | 100.00 | 61.57 | 99.07 | 99.89 | 30.53 | 98.26 | 81.94 |

Models parameterized with neural networks contribute to the inductive biases of the deep generative models (Zhang et al., 2016; Keskar et al., 2017), thus the quality of the DAG encoder can be characterized by the reconstruction accuracy (Accuracy) and the generation performance (i.e. the proportions of valid/ unique/ novel architectures in generated DAGs.) of the corresponding VAE.

The reconstruction accuracy, prior validity, uniqueness and novelty are calculated in the same way as Zhang et al. (2019). Empirical results are presented in Table 5, and we take the average of these four measurements to characterize the overall performance of the deep generative model (i.e. VAE), which also measures the quality of the DAG encoder. We find that PACE performs similarly well in reconstruction accuracy, prior validity and novelty with D-VAE, DAGNN and S-VAE, while significantly improving the uniqueness. Hence, PACE achieves the best overall performance and generates more diverse DAG architectures.

