# OpenReview forum: "PACE: A Parallelizable Computation Encoder for Directed Acyclic Graphs"
_ICLR.cc/2022/Conference — ICLR 2022 Submitted_

### Official Review · Reviewer_vSGS · 2021-10-27

**Correctness:** 3
**Technical Novelty And Significance:** 2
**Empirical Novelty And Significance:** 3
**Recommendation:** 5
**Confidence:** 5

**Main Review:**

The PACE approach proposed by the authors appears to roughly consist of the following steps:
- Solve a graph canonicalization problem using an external solver, to obtain a canonical node ordering.
- Sort the nodes into that particular ordering.
- Apply a graph neural network to the raw node indices of this node ordering for a single step. Use the resulting node embeddings as "positional embeddings".
- Construct an attention mask such that each node can only attend to its predecessors.
- Run a transformer on the new sequence using this attention mask.

My main concern with the paper is that the authors motivate their approach as a way to encode long-range dependencies in DAGs in parallel, but do not have any discussion of prior work on encoding directed-but-not-necessarily-acyclic graphs, or prior work on applying transformers to graphs. These seem like very important baselines to consider, but the authors only compare with RNN models for DAGs and a selection of undirected GNN models. Notably missing baselines and citations of related work include:

- GNN models that utilize directed edges, such as
  - Gated Graph Neural Networks [(Li et al, 2016)](https://arxiv.org/abs/1511.05493)
  - the directed variant of Message Passing Neural Networks [(Gilmer et al. 2017)](https://arxiv.org/abs/1704.01212)
  - some of the other approaches discussed in ["Relational inductive biases, deep learning, and graph networks" (Battaglia et al., 2018)](https://arxiv.org/pdf/1806.01261.pdf)
- A variety of adaptations of parallel Transformer methods to directed and undirected graphs, such as
  - Transformers with relative attention [(Shaw et al. 2018)](https://arxiv.org/abs/1803.02155)
  - the Graph Relational Embedding Attention Transformer model [(Hellendoorn et al. 2020)](https://openreview.net/forum?id=B1lnbRNtwr)
  - the Code Transformer model [(Zügner et al. 2021)](https://arxiv.org/abs/2103.11318)
  - Graphormer [(Ying et al. 2021)](https://arxiv.org/abs/2106.05234)

Although some of these are application specific, most are quite general, and could easily be adapted for the neural architecture search tasks considered here. I also don't see any particular reason to believe the particular choices made here for PACE would lead to better performance than these prior approaches, especially the models that are based on a Transformer backbone. Also, the Transformer-based methods should already be similarly parallelizable to the PACE method, which was one of the main stated benefits of PACE over the RNN-based methods. (If some of these methods are added, I would be willing to raise my score.)

Another piece of related work is ["Novel positional encodings to enable tree-based transformers" (Shiv & Quark, 2019)](https://proceedings.neurips.cc/paper/2019/file/6e0917469214d8fbd8c517dcdc6b8dcf-Paper.pdf) which proposes a set of positional encodings for trees. This doesn't target DAGs, but is closely related to PACE in terms of defining a new positional encoding and so I think it should be discussed.

The authors describe a few techniques for training the PACE architecture before fine-tuning it on a classification task: it can be trained as a VAE over graphs, or using a mask-based objective similar to BERT. Somewhat confusingly, they the VAE "training" but the BERT version "pretraining". But as far as I can understand it, both methods are interchangeable unsupervised training objectives, which could either be used as training or as a pretraining step before fine-tuning. Am I missing something? I'm also not sure if there is a fine-tuning step for any of the experiments or not.

For the experimental results, there were a few things that weren't clear to me:
- It wasn't clear what objective was used to train all of the baselines for each of the methods. Were all of them trained according to a BERT-like objective or a VAE objective, according to the task?
- The paper states that "a sparse Gaussian Process (SGP) regression model is trained to predict the DAG performance from its encoding" for the NA and BN tasks. Is this possible for all of the baselines? If not, how were the baselines trained?
- On NAS101 and NAS301, what is the actual training procedure? Is there some downstream neural network that outputs an accuracy estimate as a regression problem? Or is the model used as a generative model in some way?

I think there should be at least one experiment where the methods are all trained with a single objective and then evaluated on that objective, to determine how easily the models can learn. As it is, it seems like the models are trained using an unsupervised objective and then evaluated with some downstream task without being trained directly on that task? Although perhaps the NAS101 and NAS301 tasks involve fine tuning, it is very difficult to tell from the paper.

Something else that is likely worth discussion: At least for training the VAE, the canonical ordering appears to be used both for computing positional embeddings as well as for determining the decoding order. It seems important to disentangle these two properties, since the ordering of the output can have a significant impact on the accuracy of sequence models (see for instance [Vinyals et al. (2016)](https://arxiv.org/pdf/1511.06391.pdf)). For the "no dag2seq" ablation experiment, was the canonical ordering still used for decoding, or was it removed from both the encoding and the decoding? It would be good to have results for both conditions, to determine whether the drop in accuracy is because of the input encoding or because of the output order. (Also, what happens in the "Mask & no dag2seq" setting)?

Minor comment:
- On page 4, it states that the Agg and Combine functions are the same as in the GIN paper. However, there are multiple different things called "Aggregate" in the the GIN paper, and it is ambiguous which one you mean. Also, in GIN the inputs are vectors, not integers. It would be better to write out exactly what definitions you mean. (I think there is also a typo, in that the Agg function should take a set, like $Agg(\\{\pi(i) | (i,j) \in E \\})$, but it is written as taking two arguments instead.)

---

**Edit (Nov 23):** I have raised my score from 3 to 5 in light of the new experimental results, which address my main concern with the original paper. See my comment below.


**Summary Of The Paper:**

This paper introduces PACE, a method for converting a labeled directed acyclic graph into a sequence so that it can be fed to a Transformer architecture, along with a small modification to the Transformer to incorporate DAG structure. The authors cast their approach as a parallelizable alternative to some existing methods for encoding DAGs, which include GRU based approaches and undirected GNN based approaches, and show that their method outperforms those techniques on neural architecture search tasks. (However, the authors do not discuss or compare with existing methods for encoding of directed but not-necessarily-acyclic graphs, including other parallelizable approaches using Transformers.)

*Edit: Comparisons to other methods have been added in replies to other reviewers, although the paper has not been updated.*

**Summary Of The Review:**

The paper misses a significant body of related work on directed graphs and graph adaptations of Transformers, which would be necessary for a fair evaluation of their approach. The specific architecture choice seem fine but not especially novel or well motivated, and there are many missing details regarding the experimental results.

**Edit (Nov 23):** The authors have provided experimental results for other Transformer-based graph models, and PACE still shows strong performance on their tasks. Many of my questions about missing details have been answered in the discussion, although it is difficult to evaluate this fully without a revised version of the paper.

---

> ### Author Response · Authors · 2021-11-23
> **Response to Reviewer vSGS 1/2**
>
> Q1: Baselines (citations), such as general-purpose DAG Transformers and GNNs utilizing directed edges, are missed.
>
> A1: We appreciate that reviewer vSGS lists a large body of work relevant to the DAG encoding problem, and we agree that adding more baselines will be helpful to better evaluate the contribution of the proposed PACE model. Since it’s not straightforward to generalize tree-based Transformer [19] and positional encoding method [20] to general DAGs, we compare our PACE model against 1) recent general-purpose graph Transformers (GT [11], SAN [12], Graphormer [13], GraphiT [14]), 2) GNNs (i.e. gated-GCN + PE) that utilizes directed edges [16] and graph positional encoding method [15]. As for the directed variant of Message Passing Neural Networks [21] (Gilmer et al. 2017), such a framework is utilized in DeepGMG, which is a selected baseline in the paper (see Table 1 in the paper). The experimental results are available in our response A1 to reviewer rAFV, and we find that PACE still outperforms these general-purpose graph Transformers and directed-edge-friendly GNNs. We will make sure to include these related works and the new experimental results into our revised paper.
>
>
> [11] Dwivedi, Vijay Prakash, and Xavier Bresson. "A generalization of transformer networks to graphs." arXiv preprint arXiv:2012.09699 (2020).
>
> [12] Kreuzer, Devin, et al. "Rethinking Graph Transformers with Spectral Attention." arXiv e-prints (2021): arXiv-2106.
>
> [13] Ying, Chengxuan, et al. "Do Transformers Really Perform Bad for Graph Representation?." arXiv preprint arXiv:2106.05234 (2021).
>
> [14] Mialon, Grégoire, et al. "GraphiT: Encoding Graph Structure in Transformers." arXiv preprint arXiv:2106.05667 (2021).
>
> [15] Dwivedi, V. P.; Joshi, C. K.; Laurent, T.; Bengio, Y.; and Bresson, X. 2020. Benchmarking graph neural networks. arXiv preprint arXiv:2003.00982 .
>
> [16] Xavier Bresson and Thomas Laurent. Residual gated graph convnets. arXiv preprint arXiv:1711.07553, 2017.
>
> [19] Shiv, Vighnesh, and Chris Quirk. "Novel positional encodings to enable tree-based transformers." Advances in Neural Information Processing Systems 32 (2019): 12081-12091.
>
> [20] Ahmad, Wasi, et al. "A Transformer-based Approach for Source Code Summarization." Proceedings of the 58th Annual Meeting of the Association for Computational Linguistics. 2020.
>
> [21] Gilmer, Justin, et al. "Neural message passing for quantum chemistry." International conference on machine learning. PMLR, 2017.
>
> Q2: When training PACE, the paper uses "training" in VAE architecture but "pretraining" in the BERT version, which is somewhat confusing.
>
> A2: Both methods (i.e. "training" in VAE and "pretraining" in the BERT version) are optimizing some unsupervised training objectives before fine-tuning. After that, the trained ( pre-trained) PACE is used to encode DAGs. Then, Bayesian regression models are used to predict the performance of DAGs based on DAG encodings learnt by trained (pre-trained) PACE mode, and Bayesian optimization algorithms are used to perform the searching tasks. The reason we use the term “pretraining” in the BERT version is to make the term usage consistent with the original BERT model.

---

> ### Author Response · Authors · 2021-11-23
> **Response to Reviewer vSGS 2/2**
>
> Q3: Experimental details clarification.
>
> A3: Thanks for the detailed review. We address your concerns below.
>
> 1) In the paper, the selected baselines are trained according to the VAE objective. In the supplementary experiments (answer A1 to reviewer rAFV), the general graph Transformers (i.e. GraphiT, Graphomer, GT, and SAN) are trained according to the BERT-like objective on datasets NAS101 and NAS031, while the baselines in other cases are trained according to the VAE objective..
> 2) On dataset NA and BN, following the experimental setup in [3] and [10], all baselines are trained with a VAE objective. For instance, in the baseline GCN, it uses GCN as encoder and uses the same decoder as D-VAE [3]. As such, each trained baseline can encode DAGs into DAG encodings, then we can train the sparse Gaussian Process (SGP) regression model for each baseline.
> 3) On datasets NAS101 and NAS301, PACE is first trained according to the BERT-like objective. After that, PACE can map DAGs into a latent encoding space. Then the downstream Bayesian optimization algorithms (i.e. DNGO, DNGO-LS) (which combines Bayesian optimization and neural networks) are used to predict the accuracy (performance) of DAGs and to implement the searching task. The reason to do so is because [22] validates that the pre-training architecture generates embeddings without using accuracies can better preserve the local structural relationship in the latent space.
>
> [3] Thost, Veronika, and Jie Chen. "Directed Acyclic Graph Neural Networks." International Conference on Learning Representations. 2020.
>
> [10] Zhang, M., Jiang, S., Cui, Z., Garnett, R., & Chen, Y. (2019). D-VAE: A Variational Autoencoder for Directed Acyclic Graphs. Advances in Neural Information Processing Systems, 32, 1588-1600.
>
> [22] Shen Yan, Yu Zheng, Wei Ao, Xiao Zeng, and Mi Zhang. Does unsupervised architecture representa- tion learning help neural architecture search? Advances in Neural Information Processing Systems, 33, 2020.
>
>
> Q4: The effect of the canonical order on the decoder in PACE-VAE
>
> A4: Although the ordering of the output sequence can have a significant impact on the performance in the sequence-to-sequence model [23], it is not the same case for the decoder of PACE-VAE. Let $x_{1}, x_{2}, ...x_{n}$ be an output sequence in the sequence-to-sequence model, and $v_{1}, v_{2}, ...v_{n}$ an output node sequence in the decoder of PACE-VAE. Then, for any $i, j$ such that $i < j$, the sequence-to-sequence model knows that $x_{j}$ is arranged in a later position in the output sequence than $x_{i}$, in other words, $x_{j}$ is dependent on $x_{i}$. However, in the decoder of PACE-VAE, whether there is an edge between $v_{i}$ and $v_{j}$ is predicted by the decoder itself, hence, any topological order is suitable in the decoder of PACE-VAE, and the reason why we select (topological) canonical order is to facilitate the teacher forcing. For the same reason,  in the "no dag2seq" ablation experiment, we decode the output DAG according to the topological order.
>
> [23] Vinyals, Oriol, Samy Bengio, and Manjunath Kudlur. "Order matters: Sequence to sequence for sets." (2016).
>
> Q5: Minor comments
>
> A5: In the paper, the Agg function takes the same definition as the AGGREGATE function in equation 2.1 in GIN[1]. In the experiments, the GNN in dag2seq follows the same structure as graph convolutional layers in DGCNN [17], and details can be found in the answer A1 (1) to reviewer 1m8b. The input to the GNN is the one-hot encoding of $\pi(i)$, and we agree that it would be better to write it out. In the end, we will fix the typo in the Agg function as pointed out. Thank you for the suggestion.
>
> [1] Xu, Keyulu, et al. "How Powerful are Graph Neural Networks?." International Conference on Learning Representations. 2018.
>
> [17] Zhang, Muhan, et al. "An end-to-end deep learning architecture for graph classification." Thirty-Second AAAI Conference on Artificial Intelligence. 2018.

---

> ### Comment · Reviewer_vSGS · 2021-11-23
> **Updated review and additional comments**
>
> Thank you for the clarifications and additional experimental details. The new experimental results do seem to demonstrate that the PACE model has advantages over other directed graph encoding techniques. The additional details regarding the experimental setup are also helpful, and it would be good to add that information to the revised paper.
>
> Interestingly, the improvement seems strongest for computation-graph DAGs, and there is less of an improvement for the new ogbg-code2 benchmark, which suggests that the inductive biases of the PACE architecture are particularly well suited for computation graph DAGs (although as the authors note it may also be due to a lack of hyperparameter tuning for the new experiment).
>
> I agree with reviewer rAFV that the changes have definitely strengthened the paper significantly, but that it is difficult to evaluate fully without a revised version (especially for concerns about clarity and missing details in the original version). I have raised my score from 3 to 5.
>
> ---
>
> Follow up comments from the author response:
>
> > Both methods (i.e. "training" in VAE and "pretraining" in the BERT version) are optimizing some unsupervised training objectives before fine-tuning. After that, the trained (pre-trained) PACE is used to encode DAGs.
>
> I would recommend using the same word for both, to make it clear that both methods are using the same training procedure. Using "pretraining" for one and "training" for the other strongly suggests that the training procedure is different. (Personally, I would prefer using "training" or perhaps "representation learning", since "pretraining" makes me think the same weights will be fine-tuned on the downstream objective, but based on the clarifications it seems that the PACE weights are held fixed during the downstream Bayesian regression step. However, "pretraining" would also be a reasonable choice here, as long as you are consistent across both VAE and BERT methods.)
>
> > Although the ordering of the output sequence can have a significant impact on the performance in the sequence-to-sequence model [23], it is not the same case for the decoder of PACE-VAE. ... any topological order is suitable in the decoder of PACE-VAE, the reason why we select (topological) canonical order is to facilitate the teacher forcing. For the same reason, in the "no dag2seq" ablation experiment, we decode the output DAG according to the topological order.
>
> To clarify, I was concerned about exactly this issue of teacher forcing, and I think this could still have a significant impact for (ablations of) the PACE-VAE decoder. Enforcing a single canonical order during decoding (using teacher forcing) reduces the uncertainty in the model, by ensuring that it only has to assign probability to canonical sequences, whereas training on a random topological order during decoding (again using teacher forcing) would make the training task harder by forcing the model to assign equal probaility to every topological ordering. (There's also the option of selecting the best ordering according to the model, but that's probably impractical and prevents teacher forcing.)
>
> Since there are more than one topological ordering of a DAG, by "we decode the output DAG according to the topological order" I assume you mean the canonical order (the same sequence produced by "dag2seq")? In this case I think the comparison is valid, but it would be good to mention this detail in the revised paper. The (problematic) alternative would be to train the PACE model using the canonical order, but train the "no dag2seq" ablation on some arbitrary order (e.g. the default order in the dataset, or a randomly sampled order). In this second case, the results could be confounded by how easy it is to predict the canonical order v.s. the arbitrary order. (Ideally, it would be interesting to compare all three situations: PACE, "no dag2seq" with the canonical order, and "no dag2seq" with some arbitrary or random order. But if all of your decoders use the canonical order that also seems reasonable and there's no need to run the other experiment.)

---

### Official Review · Reviewer_1m8b · 2021-11-01

**Correctness:** 4
**Technical Novelty And Significance:** 3
**Empirical Novelty And Significance:** 3
**Recommendation:** 6
**Confidence:** 4

**Main Review:**

### Strengths
The objective of the paper is clearly motivated and a need for a parallelizable architecture is apparent. The introduction and background sections make the paper more accessible for researchers not focused on DAG modelling. Furthermore, the main aspect of the proposed method, the dag2seq framework, does not depend on any specific architecture, but instead provides an algorithm to transform a DAG into a sequence-like input. This is favourable since future work can improve/replace the network architecture while maintaining the dag2seq framework, giving this work a potentially higher impact than a specialized architecture. The usage of isomorphism classes is well justified and has no obvious drawbacks. The application of the Transformer architecture for encoding and decoding of the graphs seems a suitable choice as well. The experimental results confirm the benefit of the architecture not only in efficiency, but also in performance.

### Weaknesses
- The explanation of the used GNN in Section 3.1 is not fully stand-alone and misses out details. The text can benefit from extended it or referring to an appendix section that clarifies:
	- The explicit definition of the used aggregation and combination function;
	- Whether embeddings are used to map $\pi(i)$ to a feature space, and how those are learned;
	- An example of two graphs being mapped to the same isomorphism class.
- While transforming a DAG into a sequence of positional encodings is beneficial in having more flexibility in the architecture choice, it does not incorporate an inductive bias of the graph adjacency matrix as the message passing methods do. This also becomes apparent from the fact that an explicit masking strategy for introducing back some graph structure helps in its prediction. Similarly, as in NLP, relying on positional encodings can have a considerable impact on learning with data sparsity - on small datasets, the message passing algorithms (D-VAE/DAGNN) might outperform PACE. Thus, an experiment on e.g. the BN dataset with different training set sizes (e.g. 2k, 5k, 10k, 20k) is necessary to validate/analyze this aspect of the proposed architecture.
- Section 5.2 misses out a discussion of how the hyperparameters, especially network sizes, have been chosen across baselines to ensure a fair comparison. Was this done on number of parameters, computation time, etc.? Or how is a fair comparison ensured otherwise?
- Regarding the last aspect, the training time of PACE on the BN dataset in comparison to the asynchronous message passing algorithms is not as much lower as the text might suggest (35mins per epoch compared to 45/50mins). Does the Transformer architecture in PACE have more parameters than DAGNN/D-VAE? Can DAGNN/D-VAE benefit by having more parameters for the given datasets?
- A drawback of the dag2seq encoding method is that finding the canonical form of a graph can be a computationally-expensive pre-processing step. Thus, it is unclear whether PACE can actually encode an unseen graph, including the required pre-processing step, faster than an asynchronous message passing method at inference. Figure 3 seems to not include such a time comparison.

__Post-rebuttal update__: I appreciate the authors' rebuttal, and most of my questions have been sufficiently answered. I think the paper will significantly improve by incorporating these changes. In light of the fair concerns of other reviewers and the missing update of the main text, I will stay with my initial score, which was already weakly positive.

**Summary Of The Paper:**

This paper proposes an encoder architecture for directed acyclic graphs which is parallelizable in computation and thus more efficient than asynchronous message passing alternatives. The model is based on representing the graph in its canonical form, i.e. a unique representation of its isomorphism class, which is transformed into a positional encoding and combined with a Transformer-based encoder. In experiments on neural architecture searches and Bayesian networks, the proposed architecture outperforms related methods.

**Summary Of The Review:**

The proposed method shows favorable benefits compared to previous work. Overall, the strengths outweigh the drawbacks. Thus, my current recommendation is "Weak Accept".

---

> ### Author Response · Authors · 2021-11-23
> **Response to Reviewer 1m8b 1/2**
>
> Q1: Details of applied GNN of dag2seq are not very clear.
>
> A1: We agree that referring the details of the applied GNN (of dag2seq) in the experiments to an appendix section can be helpful, and appreciate the suggestion. We address the proposed concerns below:
>
> In our experiments, the GNN of dag2seq simply takes the same formulation as the graph convolution layer in [17]. Briefly, let $x_{j}$ be the input feature of node j (for instance, the one-hot encoding of $\pi(j)$), $W$ be a trainable parameter matrix, $d_{j}$ be the in-degree of node j. Then the Agg function is $a_{j} =\frac{1}{d_{j}} \sum_{(i,j)\in E} x_{i}W$ and the Combine function is $p_{j} =\frac{1}{d_{j}} x_{j}W + a_{j}$. Though the Agg function and Combine function are not strictly injective, the experimental results show that the generated positional encodings are still surprisingly effective.
> We simply used one-hot encoding of $\pi(i)$ in our experiments.
> In fact, Theorem 3.1 indicates that dag2seq can map any two DAGs in the same isomorphic class to the same sequence. Hence, any two isomorphic DAGs could be a good example. For instance, you may exchange the input label of the green node and pink node in DAG $G_{1}$ (see Figure 2 in the paper). It’s obvious that these two DAGs are isomorphic, while the sequences generated by dag2seq are the same.
>
> [17] Zhang, Muhan, et al. "An end-to-end deep learning architecture for graph classification." Thirty-Second AAAI Conference on Artificial Intelligence. 2018.
>
> Q2: Whether PACE incorporates the inductive bias of the adjacency matrix? Model performance on relatively small datasets?
>
> A2: As theorem 3.1 states, PACE incorporates the inductive bias of the adjacency through the proposed dag2seq framework as it injectively encodes (non-isomorphic) DAGs into (different) sequences. The reason why the masked attention mechanism helps the prediction is that it adjusts the attention scores according to the dependencies between nodes, and we provide further discussion in A1 to reviewer rAFV.
>
> The concern of the training set size is very insightful. In the selected datasets, NA contains approximately 19K DAGs. Hence it has relatively small training datasets, and our experimental results (Table 1 in the paper) indicates that PACE can achieve the state-of-art performance under such circumstances. For dataset BN, we sample a training set of size 10K (which is smaller than dataset NA) as suggested by reviewer 1m8b. Below are the experimental results, and it shows that PACE still achieves the best performance.
>
> ## dataset BN (training set size = 10K)
> |Model | RMSE (smaller = better) |Pearson’r (larger = better) | Kendall Tau  (larger = better)|
> | :------:| :------: | :------: | :------: |
> |PACE|0.147 +- 0.003| 0.990 +- 0.000| 0.908 +- 0.001 |
> |DAGNN|0.313 +- 0.005|0.836 +- 0.002| 0.817 +-  0.001|
> |D-VAE|0.623 +- 0.006| 0.793 +- 0.002| 0.592 +- 0.002 |
>
> Q3: How is a fair comparison ensured when comparing the computation cost?
>
> A3: The computation cost of each model is related to the reported hyper-parameter setting. In other words, each model (PACE as well as baselines) is tuned to achieve the approximately optimal predictive performance, then we compare the computation cost under such settings.

---

> ### Author Response · Authors · 2021-11-23
> **Response to Reviewer 1m8b 2/2**
>
> Q4: Why the speedup of PACE on dataset NA and BN are not as significant as on Dataset NAS101 and NAS301?  Can DAGNN/D-VAE benefit by having more parameters for the given datasets?
>
> A4: The reason why the speed-up of PACE on dataset NA and BN are not as significant as on Dataset NAS101 and NAS301 is because of the VAE architecture. As the appendix D indicates, when we train PACE in the VAE architecture, both the encoder and decoder consists of 3 masked Transformer blocks. However, each decoder Transformer block contains 2 multi-head (self) attention layers, while each encoder Transformer block only contains one multi-head (self) attention layer. As such, the speedup of PACE on dataset NA and BN are weakened because of introducing the additional multi-head (self) attention layers on the decoder side.
>
> The tested hyper-parameter setting of DAGNN/D-VAE can guarantee that they can achieve the approximately optimal predictive performance. For instance, the tested DAGNN has two layers. According to the Figure 4 in [9], DAGNN will not benefit by increasing the number of layers.
>
> [9] Thost, Veronika, and Jie Chen. "Directed Acyclic Graph Neural Networks." International Conference on Learning Representations. 2020.
>
> Q5: The graph canonicalization problem can be computational expensive. Whether the preprocessing time is included when comparing the computation cost?
>
> A5: In Figure 3, we include the computation time of finding the canonical form of DAGs. In the training phase, we only compute the canonical form of each graph once and then use the extracted canonical forms in each training epoch. In the inference phase, the computation time of PACE includes the pre-processing time (time of finding the canonical form) as well as the encoding time. The comparison of inference time on datasets NA and BN are available in the response A5 to Reviewer d1eo. The comparison of inference time on datasets NAS101 and NAS301 are available in the sub-figure (d) in Figure 3 in the paper. These results suggest that PACE can actually encode an unseen graph faster than DAGNN/D-VAE. We would like to thank you for the valuable question, and we will make it clear in our revised paper
>
> In practice, getting their canonical orders is not too difficult, thanks to the well-known graph canonicalization tools such as Nauty [18]. Empirically, such tools usually return the canonical order of a reasonable-sized graph in seconds. Theoretically, Nauty has an average time complexity of $O(n)$, and polynomial-time graph canonicalization algorithms also exist for graphs of bounded degrees.
>
> [18] McKay, Brendan D., and Adolfo Piperno. "Practical graph isomorphism, II." Journal of symbolic computation 60 (2014): 94-112.

---

### Official Review · Reviewer_rAFV · 2021-11-02

**Correctness:** 3
**Technical Novelty And Significance:** 3
**Empirical Novelty And Significance:** 3
**Recommendation:** 5
**Confidence:** 3

**Main Review:**

I have enjoyed reading this paper; it was well-written, to-the-point and it clearly highlighted what it brings to the table compared to prior art. The empirical performance which was provided also seemed convincing to me. While the novelty is somewhat limited (the paper essentially proposes a novel, specialised, PE for Transformers), I think that the area of application (DAGs) is interesting and prevalent enough for such a study to be warranted and useful.

However I also found that the paper has several shortcomings, in terms of the extents of its evaluation, contrasting with relevant prior work, and possible discussions on the model's limitations. I would invite the authors to carefully address the following, after which I would be happy to support the paper's acceptance:

- Foremost, PACE's contribution is centered on a DAG-friendly positional representation for Transformers. In doing so, however, based on the writeup and references, the authors seem unaware of the flurry of recent work on Graph Transformers, which generalise both the Transformer and its positional encoding to the (general) graph domain. Important references that the authors should discuss include: (Dwivedi and Bresson, AAAI'21 DLG), SAN (Kreuzer et al., NeurIPS'21), Graphormer (Ying et al., NeurIPS'21), GraphiT (Mialon et al., 2021). While some of the above have only recently been published, they have been available on the arXiv for a significant time prior to the ICLR deadline. Properly contrasting PACE against such proposals is, in my opinion, crucial to confirm that a DAG-specific PE is even necessary compared to the more generic ones.
- The authors position most of their evaluation setup based on DAGNN's, and they focus primarily on evaluating it in the self-supervised setting (e.g. with a VAE pipeline). However, DAGNN also evaluates their work on the ogbg-code benchmark, which seems to be missing from this paper. As ASTs are a standard application area for DAG-based processing, it would be very useful to show PACE's performance in this case.
- Lastly, and somewhat relatedly, even if we are able to fully parallelise the dataflow over a DAG-specific GNN, I would assume that for some tasks the complexity of covering the entire DAG's diameter may be unavoidable. For example, any dynamic programming task over a DAG that requires spreading information from a source vertex (e.g. finding shortest path lengths from a given source vertex). I would like to see some discussion from the authors regarding this.

**Summary Of The Paper:**

The authors propose PACE, a Transformer-based architecture for directed acyclic graphs (DAGs). By providing a DAG-specific positional encoding (using canonicalisation and GINs), the claim is that a fully-connected self-attentional model can be executed over such an input, ameliorating the requirement of following the DAG's topological structure in an RNN-like fashion. The proposed model is evaluated over neural architecture search datasets, showing improvements over relevant baselines.

**Summary Of The Review:**

In general I believe the idea is worthy of publication. However the concerns I presented, especially the fact the evaluation does not completely match DAGNN's and the relevant graph Transformer literature is not compared against, means the paper could benefit from another iteration of improvements. If the authors satisfactorily update the paper in response to my comments, I will back the paper for acceptance.

---

> ### Author Response · Authors · 2021-11-23
> **Response to Reviewer rAFV 1/2**
>
> Q1: Comparison of PACE and other general-purpose graph Transformers.
> A1: We agree that contrasting PACE against recent (general) graph Transformer (GT[11], SAN[12], Graphormer[13], GraphiT[14]) and graph positional encoding method [15] [16] is necessary for the evaluation of the contribution of our paper, and we appreciate the constructive feedback. In fact, our research (PACE) was primarily motivated by the prevalence of GRU-based DAG encoders in the domain of DAG optimization problem, and we started this research almost at the same period as these general graph Transformers. Hence, such comparison is missed when we formulate the paper. As suggested, here we perform some additional experiments to validate the strength of PACE against state-of-art (general) graph Transformers and graph positional encodings, and the experimental results are presented as following:
> ## dataset NA
> |Model | RMSE (smaller = better) |Pearson’r (larger = better) |
> | :------:| :------: | :------: |
> |PACE|0.254 +- 0.002 (best)| 0.964 +- 0.001 (best)|
> |GT|0.329 +- 0.001|0.942 +- 0.001|
> |SAN|0.311+- 0.003 |0.950 +- 0.001 |
> |Graphomer|0.352 +- 0.002 |0.936 +- 0.001 |
> |GraphiT| 0.299 +- 0.002 |0.955 +- 0.001 |
> |gated-GCN + PE |0.416 +- 0.002| 0.891 +- 0.001|
> ## dataset BN
> |Model | RMSE (smaller = better) |Pearson’r (larger = better) |
> | :------:| :------: | :------: |
> |PACE|0.115 +- 0.004 (best)| 0.994 +- 0.001 (best)|
> |GT|0.166 +- 0.003|0.987 +- 0.000|
> |SAN|0.158+- 0.005 |0.989 +- 0.001 |
> |Graphomer|0.181+- 0.004 |0.971 +- 0.001 |
> |GraphiT| 0.142 +- 0.005 |0.990 +- 0.003 |
> |gated-GCN + PE |0.311+- 0.003|0.953 +- 0.002|
> ## dataset NAS101 (Regret %: smaller = better)
> |Model | DNGO |DNGO-LS|
> | :------:| :------: | :------: |
> |PACE|0.391 +- 0.241 (second best) | 0.278 +- 0.178 (best)|
> |GT|0.573+- 0.276|0.460 +- 0.148|
> |SAN|0.386+- 0.250 |0.291 +- 0.166 |
> |Graphomer|0.429+- 0.302 |0.314 +- 0.182 |
> |GraphiT| 0.407+- 0.233 |0.307 +- 0.181 |
> |gated-GCN + PE |0.451+- 0.177| 0.428 +- 0.168|
> ## dataset NAS301 (Accuracy %: larger = better)
> |Model | DNGO |DNGO-LS|
> | :------:| :------: | :------: |
> |PACE|94.507 +- 0.165 (best) |94.547 +- 0.145 (second best)|
> |GT|94.421 +- 0.174|94.533 +- 0.139|
> |SAN|94.446+- 0.191 |94.501 +- 0.143 |
> |Graphomer|94.477+- 0.142 |94.551 +- 0.137 |
> |GraphiT| 94.482+- 0.156 |94.489 +- 0.138 |
> |gated-GCN + PE |94.461 +- 0.170|94.421  +- 0.164|
>
> On dataset NA and BN, PACE still consistently achieves the current state- of- art performance. On NAS101 and NAS301, PACE always achieves the top-2 performance : 1) When DNGO is used as the downstream search algorithm on NAS101, SAN > PACE > other baselines; 2) When DNGO-LS is used as the downstream search algorithm, Graphomer > PACE > other baselines; 3) In other cases, PACE still achieves the best results.
> In these baselines, GT, SAN, and gated-GCN + PE use Laplacian positional encodings (Laplacian PEs), which are generalizations of the positional encoding framework in the original Transformer model. Briefly, these Laplacian PEs are constructed from the k smallest non-trivial eigenvectors (eigenvalues) through the factorization of the graph Laplacian matrix. However, unlike a general graph, graph Laplacian based methods are inherently not suitable for DAGs. Hence, we treat DAGs as undirected graphs when computing the Laplacian PEs. For the same reason, these PEs fail to incorporate the inductive bias of the DAG encoding problem, and then these baselines are not competitive to PACE in the DAG-specific optimization problems. On the other hand, both GraphiT and Graphomer incorporate the (relative) position of nodes in graphs (DAGs) in the attention mechanism. The difference is that GraphiT uses graph kernels to adjust the attention scores, while Graphomer utilizes the distance of the shortest path to do so. Compared to the two methods, PACE uses the masked attention mechanism to adjust the attention scores according to the dependencies of nodes (further discussions are available in A1 to Reviewer d1eo)., PACE also incorporates the node relations through the proposed dag2seq (a GNN-based framework), hence the generated positional encodings (PEs) helps to incorporate the inductive bias of  the DAG encoding problem.
>
> In summary, these general-purpose graph Transformers all fail to incorporate the strong inductive bias of DAGs, thus are not suitable for DAG encoding and optimization tasks considered in the paper. Nevertheless, we very much agree that these discussions are needed. We will discuss these methods and add the new results into our paper.

---

> > ### Comment · Reviewer_rAFV · 2021-11-23
> > **Rebuttal acknowledged**
> >
> > I would like to thank the authors for their careful rebuttal and additional experimental results. These will strengthen the paper for sure.
> >
> > Unfortunately, I don't think the authors revised their paper on OpenReview (which would have made reviewing these changes more convenient).
> >
> > The new results on ogbg-code2, while important, also indicate a potential shortcoming of the model w.r.t. DAGNN when scaling up. I believe a more thorough hyperparameter sweep is in order.
> >
> > Further, there's a chance the authors misunderstood my question about diameter covering. My claim was that there exist problems on DAGs where it would be _necessary_ to perform a number of steps equal to the diameter, regardless of what kind of skip technique you're using.
> >
> > One example of this is computing shortest path lengths from a single source vertex in a DAG (specified by an input feature). I don't see an obvious way in which this can be reliably computed in fewer than O(diameter) steps, no matter what the computational graph is.
> >
> > Because of all of the above concerns, while I believe the authors' efforts may have pushed the paper over the bar, a little more care is needed before I can recommend acceptance with certainty. Therefore, I choose to retain my score.

---

> ### Author Response · Authors · 2021-11-23
> **Response to Reviewer rAFV 2/2**
>
> References of A1:
>
> [11] Dwivedi, Vijay Prakash, and Xavier Bresson. "A generalization of transformer networks to graphs." arXiv preprint arXiv:2012.09699 (2020).
>
> [12] Kreuzer, Devin, et al. "Rethinking Graph Transformers with Spectral Attention." arXiv e-prints (2021): arXiv-2106.
>
> [13] Ying, Chengxuan, et al. "Do Transformers Really Perform Bad for Graph Representation?." arXiv preprint arXiv:2106.05234 (2021).
>
> [14] Mialon, Grégoire, et al. "GraphiT: Encoding Graph Structure in Transformers." arXiv preprint arXiv:2106.05667 (2021).
>
> [15] Dwivedi, V. P.; Joshi, C. K.; Laurent, T.; Bengio, Y.; and Bresson, X. 2020. Benchmarking graph neural networks. arXiv preprint arXiv:2003.00982 .
>
> [16] Xavier Bresson and Thomas Laurent. Residual gated graph convnets. arXiv preprint arXiv:1711.07553, 2017.
>
>
> Q2: Performance of PACE on ogbg-code2 (application of AST).
>
> A2: We implement PACE on ogbg-code2 to evaluate its power on the task of encoding ASTs. The experimental results suggest that PACE achieves comparable performance with other state-of-art DAG encoders, while significantly reducing the (training and) inference time. The details of the experimental results can be found in A2 to the reviewer d1eo.
>
> Q3: Discussion of the complexity of covering the entire DAG’s diameter.
>
> A3: Thanks for the insightful question. In PACE, the GNN in dag2seq framework is designed to encode the interactions between nodes in the input DAG, and it is not required to cover the diameter. For the same reason, the GNN only takes a single layer, and the following experimental results (on dataset BN) show that PACE will not benefit from increasing the number of layers in GNN (in dag2seq framework). As for the diameter consideration, the (masked) Transformer layer will propagate information of all connected nodes in the DAGs to the center node, hence it covers the entire diameter.
>
> ## Dataset BN:
> |Methods | RMSE | Pearson’r |
> | :------:| :------: |  :------: |
> |PACE (1-layer GNN)| 0.115 | 0.994|
> |PACE ( 2-layer GNN)| 0.118| 0.994|

---

### Official Review · Reviewer_d1eo · 2021-11-02

**Correctness:** 3
**Technical Novelty And Significance:** 2
**Empirical Novelty And Significance:** 3
**Recommendation:** 5
**Confidence:** 4

**Main Review:**


Positive points:
1.	Well-structured presentation with good figure illustration of the core problem and proposed framework.
2.	Accelerating the encoding of DAGs is of great importance and the proposed solution seems to be novel with convincing results.

Concerns:
1.	Not enough technical motivations for creating such a structure. I can understand why the structure could make it parallelizable, however, why we choose such as combination of the dag2set + masted transformer is still not well-presented.
2.	Dataset with larger graphs (graph with more nodes) is needed to demonstrate the performance. Also, better downstream tasks should be added for validating the proposed method: (1) it’s better to combine the proposed method with advanced NAS baselines, the BO-based method is well-recognized in the AutoML domain but not for the latest NAS community. (2) it’s better to conduct other task for validating the graph encoding model such as protein classification, computer program encoding, etc. NAS task itself could be of too much variance to validate the performance.
3.	RMSE and Pearson correlation may not be good metric for validating the performance even compared to non-parametric ranking metrics such as Kendall Tau ranking coefficients.
4.	Though comparing the training speed is good to show the general speed of learning the models, but I don’t think it’s good to show the benefit of parallel encoding of the nodes compared to using the inference speed.


**Summary Of The Paper:**

The paper presents a new architecture for encoding the directed acyclic graphs into embedding vectors to benefit the downstream applications. The proposed architecture breaks the limitation of sequentially encoding the nodes of a DAG, and could parallelly encode the nodes of each graph so that the learning and inference speed could be much improved. The proposed method is validated on encoding neural architectures collected in the NAS benchmarks and shows better effectiveness and efficiency compared to existing graph encoding methods.

**Summary Of The Review:**

Based on the above concern described above, I currently tend to weakly reject the paper but would like to hear more feedback from the author and discussion from other reviewers towards the final decision.

---

> ### Author Response · Authors · 2021-11-23
> **Response to Reviewer d1eo 1/2**
>
> Q1: Technical motivation of the structure design (dag2seq + masked self-attention) is not clear.
>
> A1: Currently, the state-of-art DAG encoders (i.e. D-VAE, DAGNN) are GRU-based, hence they inheritthe limitations of RNN models, such as the slow training (inference) speed, difficulty to capture the long-term dependency of nodes, etc. In order to overcome these limitations, we propose to borrow the power of Transformer for sequence modeling to the DAG modeling problem. In order to do so, it’s intuitive to injectively represent non-isomorphic DAGs as different sequences. However, as we discussed in section 2.2 of the paper, some naive approaches are ambiguous. Hence, we borrow ideas from GIN [1] and practical graph canonicalization tools [2] (such as Nauty) to design the dag2seq, which provides a way to injectively encode DAGs into sequences (Theorem 3.1 in the paper). Furthermore, since any two items in a sequence are correlated with each other, a standard Transformer performs the full-sequence attention. On the contrary, for node pairs $(v_{i}, v_{j})$ in a DAG, they can be categorized into two different groups by checking whether there is an directed path from node $(v_{i}$ to node $(v_{j}$, and the dependency between nodes exists only when the directed path exists. Hence, PACE chooses to use masked attention to distinguish node pairs, which helps to learn the dependency between nodes better.
>
> [1] Xu, Keyulu, et al. "How Powerful are Graph Neural Networks?." International Conference on Learning Representations. 2018.
>
> [2] McKay, Brendan D., and Adolfo Piperno. "Practical graph isomorphism, II." Journal of symbolic computation 60 (2014): 94-112.
>
> Q2: Performance on datasets with larger graph (ogbg-code) .
>
> A2: Our paper focuses on the DAG optimization problem. Since searching the optimal neural architectures and Bayesian network structures are essentially typical DAG optimization tasks, we choose datasets NA, NAS101, NAS301 (neural architecture search benchmark) and BN (Bayesian network structure learning benchmark) for the purpose of evaluation. However, as mentioned by Reviewer d1eo and reviewer rAFV, ASTs (Abstract Syntax Trees) can be another application area, here we implement additional experiments on ogbg-code2 benchmark.  Dataset ogbg-code2 contains relative large-scale graphs. On average, DAGs in the ogbg-code contain 125 nodes, and the largest DAG contains more than 30,000 nodes. Following the experimental setup of DAGNN [3], we perform the token prediction task (TOK) and use the F1 score as the evaluation metric. In the experiment, we do 5 independent trials and then take the average. We find that PACE still significantly reduces the inference time compared to GRU-based DAG encoders (i.e. D-VAE and DAGNN). However, PACE’s F1 score is not better than the baselines. Though we do not have enough time to tune hyperparameters of PACE (we will update the results if hyperparameter tuning significantly improves the result), another potential reason is that DAGNN and D-VAE follow the GNN neighborhood aggregation architecture (i.e. aggregate and combine). As the neighborhood aggregation architecture essentially learns the (sub-)tree structures, D-VAE and DAGNN can benefit from such property and henceforth are more effective for encoding abstract syntax trees (ASTs). Another possible reason is that the current PACE doesn’t use the edge features, which may cause information loss and weaken the performance.
>
> |Model |Ave F1 score (lager = better) |Ave training time (min) / epoch|
> | :------:| :------: | :------: |
> |PACE|0.1593|67|
> |D-VAE|0.1591|131|
> |DAGNN|0.1751|138|
>
> [3] Thost, Veronika, and Jie Chen. "Directed Acyclic Graph Neural Networks." International Conference on Learning Representations. 2020.

---

> ### Author Response · Authors · 2021-11-23
> **Response to Reviewer d1eo 2/2**
>
> Q3: Downstream evaluation tasks choices.
>
> A3: In NAS community, besides the DAG encoding algorithm, the model performance is heavily dependent on the downstream search algorithm (e.g. local search [3], reinforcement learning [4], Bayesian optimization [5], neural predictor [6], weight-sharing methods [7], etc). Thus it’s hard to test all these downstream methods. However, a DAG encoder that generates smooth latent search space (DAG encoding space) and preserves the strong locality of neural architectures always leads to better downstream searching performance [8]. Therefore, this paper aims to independently evaluate a model’s DAG encoding ability without resorting to search-algorithm-dependent NAS performances. We choose to use Bayesian optimization methods because the Gaussian process used in BO can exactly be used to characterize the smoothness of the learnt DAG encodings by giving a nonparametric estimation of the performance surface. Hence, we take two popular Bayesian optimization algorithms as our downstream algorithms. To reflect the variance in the evaluation process on NAS datasets (i.e. NAS101 and NAS301), we perform 50 independent runs and report the variance, and our results (Table 3 in paper) indicates that the experimental results are robust.
>
> [3] Ottelander, T. D., Dushatskiy, A., Virgolin, M., and Bosman, P. A. Local search is a remarkably strong baseline for neural architecture search. In arXiv:2004.08996, 2020.
>
> [4] Tan, M., Chen, B., Pang, R., Vasudevan, V., Sandler, M., Howard, A., and Le, Q. V. Mnasnet: Platform-aware neural architecture search for mobile. In CVPR, 2019.
>
> [5] Zhou, H., Yang, M., Wang, J., and Pan, W. BayesNAS: A Bayesian approach for neural architecture search. In ICML, 2019.
>
> [6] Yan, S., Zheng, Y., Ao, W., Zeng, X., and Zhang, M. Does unsupervised architecture representation learning help neural architecture search? In NeurIPS, 2020.
>
> [7] Pham, H., Guan, M. Y., Zoph, B., Le, Q. V., and Dean, J. Efficient neural architecture search via parameter sharing. In ICML, 2018.
>
> [8] Choi, K., Choe, M., and Lee, H. Pretraining neural architec- ture search controllers with locality-based self-supervised learning. In arXiv preprint arXiv: 2103.08157, 2021.
>
>
> Q4: Evaluation metrics (RMSE and Pearson correlation) for datasets NA and BN.
>
> A4: We used RMSE and Pearson correlation as the evaluation metrics following previous works [9,10]. However, as the reviewer suggested, here we also use the Kendall Tau ranking coefficient as an evaluation metric. As the results show (First Table: NA; Second Table: BN), we get the same conclusion regardless of the choice of the evaluation metric (RMSE, Pearson’r or Kendall Tau).
>
> ## dataset NA
> |Model | RMSE (smaller = better) |Pearson’r (larger = better) | Kendall Tau  (larger = better)|
> | :------:| :------: | :------: | :------: |
> |PACE|0.254 +- 0.002| 0.964 +- 0.001| 0.826 +-0.001 |
> |DAGNN|0.264 +- 0.004|0.964 +- 0.001|0.822 +- 0.001 |
> |D-VAE|0.384 +- 0.002|0.920 +- 0.001 | 0.809 +- 0.002|
>
> ## dataset BN
> |Model | RMSE (smaller = better) |Pearson’r (larger = better) | Kendall Tau  (larger = better)|
> | :------:| :------: | :------: | :------: |
> |PACE|0.115 +- 0.004| 0.994 +- 0.001|  0.927 +- 0.001|
> |DAGNN|0.122 +- 0.004|0.993 +- 0.000|  0.915 +- 0.001|
> |D-VAE|0.281 +- 0.004|0.964 +- 0.001 | 0.832 +- 0.002|
>
> [9] Thost, Veronika, and Jie Chen. "Directed Acyclic Graph Neural Networks." International Conference on Learning Representations. 2020.
> [10] Zhang, M., Jiang, S., Cui, Z., Garnett, R., & Chen, Y. (2019). D-VAE: A Variational Autoencoder for Directed Acyclic Graphs. Advances in Neural Information Processing Systems, 32, 1588-1600.
>
>
> Q5: Inference speed comparison on datasets NA and BN.
>
> A5: Thanks for the suggestion. Here we compare the average inference time (min) per epoch on datasets NA and BN.
>
> ## Inference time comparison (min/epoch)
> |Model | NA | BN |
> | :------:| :------: | :------: |
> |PACE|1.4|21|
> |D-VAE|2.7|132|
> |DAGNN|3.0|36|

---

### Author Response · Authors · 2021-11-23
**Meta Response to all reviewers**

We thank all the reviewers and appreciate their constructive feedback. These comments will be valuable for improving our manuscript.

1. The major common concern of reviewers is the lack of comparison and citation of necessary baselines, including recent general-purpose graph Transformers (GT [11], SAN [12], Graphormer [13], GraphiT [14]) and directed-edge-friendly GNNs (gated-GCN[16] + Laplacian PE[15]). Hence, we evaluate PACE against the baselines in the rebuttal session, and experimental results are available in our response A1 to reviewer rAFV. In summary, PACE still outperforms the additional baselines, and we will include these related works and experimental results in our revised paper.

2. In addition, reviewers also suggest evaluating PACE on ogbg-code2 benchmark. We include the experimental result in the response A2 to reviewer d1eo, and find that PACE can achieve comparable performance with state-of-art DAG encoders (D-VAE, DAGNN), while significantly boosting the training/inference time.

3. In the end, reviewers propose comments related to the model/experiment/evaluation choices, and we address the concerns with necessary additional experiments and make clarifications in the according response to each reviewer. Specifically, (1) the experiment in the answer A4 to reviewer d1eo provides results that uses Kendall Tau ranking coefficient as evaluation metric; (2) The experiment in the answer A3 to reviewer rAFV shows that a single layer GNN in dag2seq is enough and the (masked) Transformer layer will cover the entire DAGs’ diameter; (3) The experiment in the answer A2 to reviewer 1m8b shows that PACE is effective even if the training set is relatively small.

[11] Dwivedi, Vijay Prakash, and Xavier Bresson. "A generalization of transformer networks to graphs." arXiv preprint arXiv:2012.09699 (2020).

[12] Kreuzer, Devin, et al. "Rethinking Graph Transformers with Spectral Attention." arXiv e-prints (2021): arXiv-2106.

[13] Ying, Chengxuan, et al. "Do Transformers Really Perform Bad for Graph Representation?." arXiv preprint arXiv:2106.05234 (2021).

[14] Mialon, Grégoire, et al. "GraphiT: Encoding Graph Structure in Transformers." arXiv preprint arXiv:2106.05667 (2021).

[15] Dwivedi, V. P.; Joshi, C. K.; Laurent, T.; Bengio, Y.; and Bresson, X. 2020. Benchmarking graph neural networks. arXiv preprint arXiv:2003.00982 .

[16] Xavier Bresson and Thomas Laurent. Residual gated graph convnets. arXiv preprint arXiv:1711.07553, 2017.

---

### Author Response · Authors · 2021-11-24
**Clarification of the revised paper**

We appreciate the careful review and valuable additional feedback of all reviewers. Since it's my first time submit a paper to ICLR, I did not know that we can upload the revised paper during the rebuttal session, and thought that we can only upload our responses to reviewers as NeurIPS2021 during the rebuttal session. Here, we promise that we will polish the revision and add all changes in the response into the revision. We are sorry for the inconvenience.

---

### Decision · Program_Chairs · 2022-01-20

**Decision:**

Reject

**Comment:**

Thank you for your first (hopefully of many!) submissions to ICLR.
This work describes a method for allowing nodes to be processed concurrently instead of sequentially, allowing for a reduction in computation time.
The reviewers identified a number of concerns about the paper (lack of citations and baselines, an additional experiments demonstrating scale, and a number of clarifications and motivation in the text). The authors addressed the majority of these concerns due the rebuttal. I'm afraid a promise of a revised manuscript is not a sufficient substitute for the reviewers seeing a revised manuscript, and due the nature of the feedback, a revision is needed, which the reviewers have not seen to check their concerns are fully addressed. Therefore, at this stage, unfortunately, I recommend rejection.